# THE SERIAL SCALING HYPOTHESIS

**Yuxi Liu**[1][*]  **Konpat Preechakul**[1][*]  **Kananart Kuwaranancharoen**[2]  **Yutong Bai**[1]

[1]UC Berkeley  [2]Independent Researcher

## ABSTRACT

While machine learning has advanced through massive parallelization, we identify a critical blind spot: some problems are fundamentally sequential. These "inherently serial" problems—from mathematical reasoning to physical simulations to sequential decision-making—require sequentially dependent computational steps that cannot be efficiently parallelized. We formalize this distinction in complexity theory, and demonstrate that current parallel-centric architectures face fundamental limitations on such tasks. Then, we show for first time that diffusion models despite their sequential nature are incapable of solving inherently serial problems. We argue that recognizing the serial nature of computation holds profound implications on machine learning, model design, and hardware development.

serial-scaling-hypothesis.github.io

## 1 INTRODUCTION

The scaling up of machine learning has driven remarkable progress (Achiam et al., 2023; Hoffmann et al., 2022; Dosovitskiy et al., 2021; Kaplan et al., 2020; Krizhevsky et al., 2012), much of this has come from parallel scaling: hardware shifted from CPUs to massively parallel GPUs, architectures moved from RNNs to highly parallelizable Transformers, and algorithms increasingly exploit parallel compute (Dao, 2024; Chowdhery et al., 2023; Jouppi et al., 2017). But some problems stubbornly resist such advances (Aggarwal & Welleck, 2025; Kazemi et al., 2025; Li et al., 2024; Merrill & Sabharwal, 2023b; Marcus, 2018; Chollet, 2019): for a class of tasks, only scaling serial computation—allowing models to perform more sequential steps—yields further progress. It seems not all scaling is created equal.

The necessity of deep or sequential models for some problems has remained mostly theoretical (Telgarsky, 2016; Merrill & Sabharwal, 2025; Chen et al., 2024a). Only recently has scaling test-time computation become recognized, independently of train-time computation (Snell et al., 2024; OpenAI, 2024; Wu et al., 2025). Yet this dichotomy still overlooks the role of parallel vs. serial computation, which is central to this paper. The literature on scaling still commonly reports both parameter counts and computation FLOPs each as a single number, treating width (parallel) and depth (serial computation) interchangeably (Kaplan et al., 2020; Hoffmann et al., 2022; OpenAI, 2024; Snell et al., 2024).

Consider Sudoku—a number-placement puzzle requiring each number 1–9 to appear once per row, column, and subgrid—as a parable (Figure 1). Easy puzzles can be solved by filling in many blanks independently, in parallel. Hard puzzles, however, require a long chain of dependent reasoning: each blank depends on the others. No algorithm can shortcut the process.

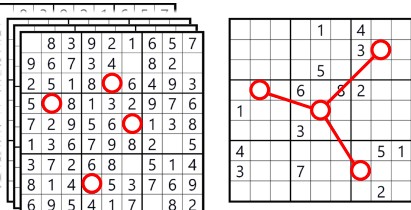

Solving many easy puzzles may cost the same total computation as solving a single hard one, but only the easy ones can be sped up with more processors. This distinction—between problems that are "wide" (parallel) and those that are "deep" (inherently serial)—is fundamental, yet it is underappreciated in machine learning.

Figure 1: (Left) Many easy Sudoku puzzles, where the circled blanks can be filled independently in parallel. (Right) A hard Sudoku with the same total compute, but the circled blanks are interdependent, requiring sequential reasoning.

---

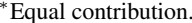

[*]Equal contribution.

> **The Serial Scaling Hypothesis (SSH)**
>
> For many important ML problems such as reasoning, decision making, and modeling dynamic systems, increasing parallel computation alone is insufficient. Progress requires scaling the amount of serial computation.

The appeal of this hypothesis is that it is both theoretically sound and practically relevant:

- **Grounded in theory:** Complexity theory proves some problems parallelize efficiently, while others do not (Greenlaw et al., 1995).

- **Explains past successes:** The breakthrough of deep learning came from increasing network depth (LeCun et al., 2015; Prince, 2023), and Chain-of-Thought (CoT) improves performance by adding more serial steps (Kojima et al., 2022; Li et al., 2024; Merrill & Sabharwal, 2024).

- **Connects to practice:** The parallel–serial lens connects well to practice. Formal results show bounded serial capacity in Transformers (Merrill & Sabharwal, 2023b) and expanded capacity with CoT (Li et al., 2024; Merrill & Sabharwal, 2024), matching empirical gains (Aggarwal & Welleck, 2025; Muennighoff et al., 2025). **Our new analysis** extends this picture to diffusion: despite thousands of iterations, diffusion models with a $\mathsf{TC}^0$ backbone have only constant computation depth, consistent with observed step-count plateaus (Ma et al., 2025; Ravishankar et al., 2024).

**Why does this matter for machine learning?** As we push toward more challenging tasks—advanced reasoning, physical simulations, planning, and scientific discovery—we encounter problems that parallel architectures (like Transformers and diffusion models) cannot efficiently solve. Recognizing this has several implications:

- Model: Should we revisit architectures that allow for deeper or more sequential computation?

- Hardware: Is it time to invest in faster, lower-latency processors, not just more parallel ones?

- Task analysis: Can some inherently serial tasks be reformulated into other tasks with reduced serial structure, while remaining useful in practice?

**Contributions.** We (i) introduce the Serial Scaling Hypothesis (SSH), (ii) prove that diffusion models, despite many iterative steps, have limited serial capacity, (iii) prove that certain Markov decision problems require serial computation for good decisions, (iv) identify machine learning problems where serial computation is essential, and (v) discuss SSH implications for machine learning.

## 2 Machine Learning from the Serial Perspective

We first formalize inherently serial problems (Section 2.1), show they are not only theoretically valid but also pervasive in machine learning (Section 2.2). We then examine why predominant ML models struggle with these tasks (Section 2.3). Finally, we discuss broader implications for ML (Section 2.4).

### 2.1 Complexity Framework for Inherently Serial Problems

We focus our attention on *binary decision problems* which take in $N$ input tokens and output either "yes" or "no" as depicted in Figure 2(A). While seemingly limited, this framework includes all problems with discrete responses, since making a discrete choice out of $2^n$ options is equivalent to answering $n$ yes/no questions. Figure 2 (C–E) show how cellular automata, many-body mechanics, and open-ended question answering can be represented in this way.

We adopt the complexity class $\mathsf{TC}$ (**T**hreshold **C**ircuits) to formally distinguish between serial and parallel problems, since it is the standard theoretical framework that formalizes the serial–parallel divide, and it has strong connections to neural networks.

**Definition 2.1** (Informal, see Appendix D)**.** A problem is in $\mathsf{TC}^i$ if, for inputs of size $N$, it can be solved by a Boolean circuit with polynomial width and polylogarithmic depth, using basic logic gates (AND, OR, NOT) plus majority gates. The class $\mathsf{TC}$ is the union of all such $\mathsf{TC}^i$ for $i \geq 0$.

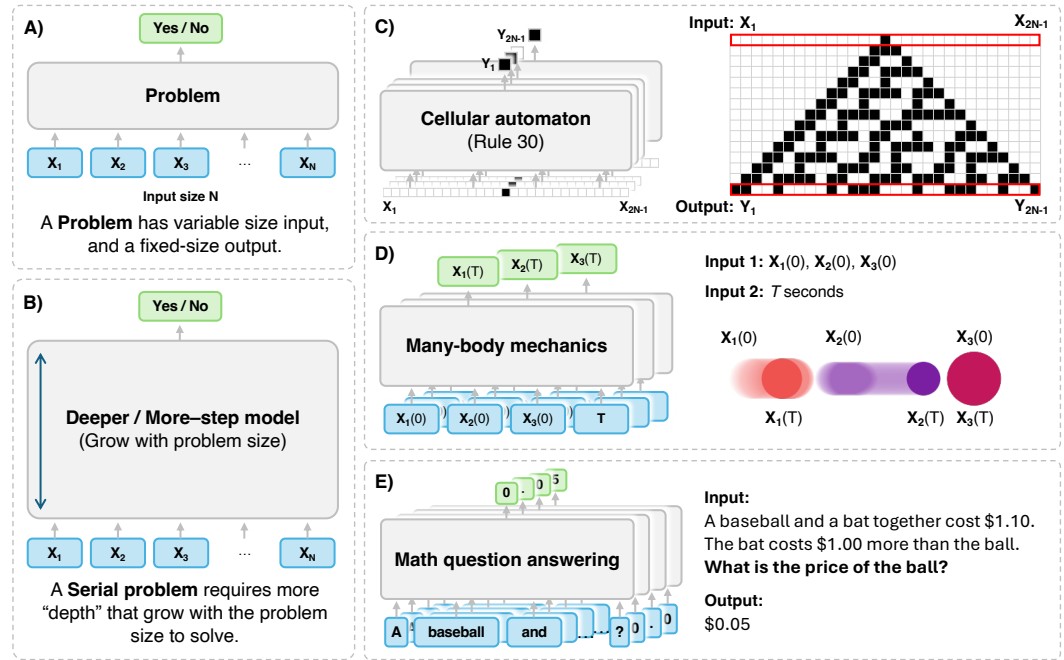

Figure 2: . (A) A **decision problem** has a variable-size input and a fixed-size output (e.g., "yes"/"no"). (B) A **serial problem** requires deeper or more steps as the problem size grows. Examples of serial problems are: (C) **Cellular automaton**: takes the initial state as input and outputs a discrete value of the row $N$ at cell $i$ for $i \in \{1, \ldots, 2N - 1\}$. (D) **Many-body mechanics**: takes initial positions and momenta of each particle with time $T$ as inputs and outputs the particle locations at time $T$ in a limited-precision space. (E) **Math QA**: takes a question as input and outputs the answer autoregressively, with each output from a fixed set of possibilities.

Intuitively, a problem belongs to $\mathsf{TC}$ if and only if it can be solved by a multilayer perceptron (MLP) with *polynomial width and polylog depth*, where polylog denotes $\mathrm{poly}(\log N)$ (Parberry & Schnitger, 1988). A problem belongs to $\mathsf{TC}^0$ if and only if it can be solved by a constant-depth MLP. We consider $\mathsf{TC}$ problems as *parallel* problems, since there are sublinear-depth MLPs that solve them.

Although the universal approximation theorem (Cybenko, 1989; Hornik et al., 1989) shows that a 3-layer MLP with unbounded width can approximate any continuous function, a constant-depth MLP with only polynomial width is far more limited, capturing only problems in $\mathsf{TC}^0$.

A common phenomenon in computational complexity is that one can make a neural network shallower at the cost of exponentially larger width (Valiant, 1983; Hajnal et al., 1993; Sherstov, 2007; Williams, 2014; Oliveira & Santhanam, 2015; Eldan & Shamir, 2016; Liang & Srikant, 2016; Cohen et al., 2016; Telgarsky, 2016; Merrill & Sabharwal, 2025). However, such exponential-sized networks are intractable in both memory and computation. We consider them as *inefficient* solutions in this paper. This exponential depth–width trade-off underlines the importance of characterizing neural networks *depth* and *width* separately, which motivates the Serial Scaling Hypothesis.

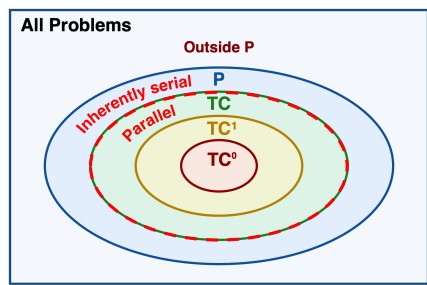

Figure 3: The complexity classes are nested as $\mathsf{TC}^0 \subseteq \mathsf{TC}^1 \subseteq \cdots \subseteq \mathsf{TC} \subseteq \mathsf{P}$. Each containment is widely believed to be strict. Problems in $\mathsf{TC}$ are parallel, while those outside are inherently serial.

To formally characterize the problems in terms of depth, we have the following definition:

**Definition 2.2.** A problem $\mathcal{P}$ is **parallel** if $\mathcal{P} \in \mathsf{TC}$; otherwise, it is **inherently serial**.

**Assumption 2.3.** We adopt the widely held belief that $\mathsf{TC} \subsetneq \mathsf{P}$—that is, some polynomial-time problems are inherently serial (Greenlaw et al., 1995, Ch. 5, Ch. 8).

## 2.2 REAL-WORLD PROBLEMS ARE LIKELY INHERENTLY SERIAL

A natural objection to theory: "Do such theoretical obstructions to parallelism arise in the real world?" We say: Yes—particularly if one works in domains involving the need for physical simulation or reinforcement learning. But even outside these areas, it remains highly likely that one will encounter inherently serial problems. Even apparently simple tasks with a polynomial-time algorithm can be inherently serial (Assumption 2.3).

Drawing from both empirical observations and theoretical arguments presented in the following Section 3, we propose the following complexity-theoretic hypothesis:

> **Hypothesis 2.4**
>
> The following problems—cellular automata evolution, many-body mechanics, sequential decision problems, mathematical question answering—are inherently serial.

Inherently serial problems (see Definitions 2.1, 2.2) exhibit fundamental computational dependencies: the outcome of intermediate steps directly influences subsequent steps in ways that cannot be shortcut without compromising correctness. To make this intuition rigorous, we prove a new theorem (Theorem G.2) on inherent seriality in sequential decision making.

**Limitations.** (1) These arguments apply to the general (worst-case) complexity of the problems. A problem may be inherently serial in general, but a parallel algorithm may cover most specific instances occurring in practice. (2) An inherently serial problem may be solved approximately in parallel to an accuracy acceptable for practical use.

## 2.3 ARE OUR MODELS CAPABLE OF SOLVING INHERENTLY SERIAL PROBLEMS?

Most modern architectures are designed for scaling parallel computation. Transformers and state-space models (SSMs) (Gu et al., 2021; Gu & Dao, 2023), though often called "sequence" models, process all available input in parallel. This raises a crucial question: "How efficiently do they solve inherently serial problems?"

Prior work shows that for fixed-depth, polynomial width, and fixed-precision, various architectures, including MLPs, Transformers, and linear SSMs, all collapse into constant-depth threshold circuits ($TC^0$) (Merrill et al., 2022; Chiang, 2024; Chen et al., 2024b; Merrill et al., 2024). Intuitively, their computational graphs all collapse into constant parallel steps, never requiring long sequential chains. Importantly, these results characterize the *architectures themselves*. A *model*, however, is the combination of both architecture and inference procedure. Thus, while a transformer architecture lies in $TC^0$, a Transformer *model* run with serial inference—such as autoregressive CoT or recurrence—extends beyond $TC^0$ and can capture inherently serial computations (see Appendix C.1). Table 1 summarizes computational characteristics of common machine learning models.

| Method | Parallel? | Solve serial problem? |
|---|---|---|
| FF MLPs | ✓ | ✗ |
| FF Transformers | ✓ | ✗ |
| FF SSMs (Mamba) | ✓ | ✗ |
| RNNs | ✗ | ✓ |
| Repeating layers | ✗ | ✓ |
| Chain-of-Thought | ✗ | ✓ |
| Diffusion models ($TC^0$ backbone) | ✗[1] | ✗ |

Table 1: Parallelizable models are limited to parallel problems. Only non-parallelizable models may solve inherently serial problems. FF stands for Feedforward.

**Diffusion models cannot solve inherently serial problems.** A key contribution of the present paper is a new theorem (Theorem F.2) on the limitations of diffusion modeling. Although the step-by-step sampling of a diffusion model appear serial, we prove in Section 4 that a diffusion model with a $TC^0$ backbone remains in $TC^0$, even with infinitely many sampling steps. Thus, despite their stepwise structure, diffusion models are incapable of solving inherently serial problems.

By linking the computational limits of modern ML models with the inherently serial structure of many key problems, we arrive at the following conclusion:

---

[1]While diffusion models are not yet parallelizable, given their $TC^0$ capabilities, there might exist a parallelization algorithm for diffusion models.

> **Key Limitation of Modern Machine Learning Models**
>
> MLPs, Transformers, SSMs, and diffusion models with $TC^0$ backbones are provably incapable of solving general instances of inherently serial problems such as cellular automata evolution, many-body mechanics, sequential decision-making, and mathematical question answering.

**Limitations.** (1) Certain instances of serial problems may be trivial or admit parallel solutions; seriality holds only in the general case. (2) For problems beyond P (e.g., NP \ P), exponential computation—not serial computation—dominates. Thus, the Serial Scaling Hypothesis applies most directly to problems of practical real-world difficulty.

## 2.4 IMPLICATIONS OF THE SERIAL SCALING HYPOTHESIS

**For machine learning practitioners.** Many important real-world problems—such as cellular automata, many-body mechanics, and sequential decision-making—are inherently serial. Without sufficient serial computation, solving them with shallow models requires an exponentially large set of weights, which in turn demands exponentially large datasets to train. Neither is affordable in practice. This mismatch—*solving inherently serial problems using shallow or parallel models*—helps explain why many ML systems generalize poorly beyond their training distributions (Torralba & Efros, 2011; Zhang et al., 2017; Recht et al., 2019; Liang et al., 2023; Mancoridis et al., 2025; Zhang et al., 2025).

Evidence from RL fine-tuning (Wang et al., 2025b; Yue et al., 2025; Shao et al., 2025) suggests that LLMs already acquire both shallow heuristics (fast but memorization-like (Nikankin et al., 2025)) and deeper algorithms (slower but more general) during pretraining. However, if the inference-time compute budget is set too low, the model defaults to the shallow routines, producing fast but brittle behavior. Only with sufficient serial computation, the deeper routines can be executed resulting in better generalization.

**For task & benchmark designers.** Recognizing the inevitable cost to inherent seriality, serial problems may be reformulated into coarser or approximate problems to reduce serial depth to acceptable levels while remain practically useful. For RL, truncated value functions cap the effective depth while retaining theoretical guarantees and practical utility (Park et al., 2025; Sutton & Barto, 2018; De Asis et al., 2019). For reasoning, complexity theory shows that coarse decision problems can be tractable even when exact solutions are not—for example, primality is in P while factoring remains hard (Agrawal et al., 2004).

In addition, benchmarks should have inherently serial problems as a separate category, to distinguish serial and parallel scaling in model performance.

**For model designers.** Solving these challenging real-world problems may require recurrent structures that increase serial computation alongside today's predominantly parallel designs. However, recurrence and depth often amplify gradient variance (Bengio et al., 1994; Pascanu et al., 2013) and L-Lipschitzness (Bartlett et al., 2017; Fazlyab et al., 2019), making models harder to train. This motivates improved training techniques and novel architectures such as implicit gradients (Wang et al., 2025a), xLSTM (Beck et al., 2024), and test-time training (Sun et al., 2024).

**For hardware designers.** Massively parallel computing machinery, especially GPU clusters, enabled past progress in deep learning. Future progress in machine learning and computing in general[2] also depends on progress in high-clockrate, sequential computing machinery, with reduced data movement overhead (Kang et al., 2021; Kaur et al., 2024). A concrete example is wafer-scale hardware (e.g., Cerebras CS-3), which prioritizes extremely high on-chip memory bandwidth (Wang, 2025).

## 2.5 RELATED WORKS

Our hypothesis is similar to the "parallelism tradeoff" hypothesis—that all parallelizable models, irrespective of design, must necessarily fail to solve inherently serial problems (Merrill & Sabharwal, 2023a;b). Our work builds on classical complexity theory, including the "depth-width

---

[2]Amdahl's law (Amdahl, 1967; Gustafson, 1988) shows that seriality is a hard upper bound on speedups achievable by parallelism.

tradeoff" (Vishkin & Wigderson, 1985), the "work" vs. "depth" contrast Blelloch (1996), P-completeness (Greenlaw et al., 1995, Ch. 8), and computational irreducibility (Wolfram, 2002).

While these ideas are well-established in complexity theory, our work brings them into the machine learning context—extending them to real-world tasks such as sequential decision-making and question answering, and highlighting the gap between these tasks and the limited serial capabilities of modern machine learning models.

## 3 SERIAL PROBLEMS

In this section, we highlight representative inherently serial problems: cellular automata (Section 3.1), physical systems (Section 3.2), and complexity-theoretic hardness results (Section 3.3), as well as practical domains such as sequential decision making (Section 3.4) and reasoning QA (Section 3.5). These cases illustrate serial bottlenecks that readers may encounter in other domains.

### 3.1 CELLULAR AUTOMATA

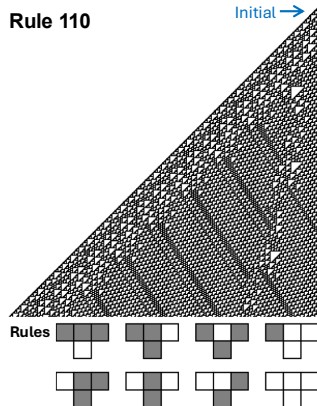

We begin with predicting the outcome of cellular automata (CA) (Sarkar, 2000; Codd, 2014), shown in Figure 2(C), as a simple problem that is inherently serial. A CA is a grid of cells, each in one of finitely many states, updated in discrete steps according to a local rule based on its neighbors. Despite this simplicity, CA can exhibit behaviors ranging from predictable to complex dynamics such as Rule 110 (Figure 4), which hints at inherent seriality of this problem.

**Cellular automata are inherently serial.** Rule 110 has been proven Turing-complete (Woods & Neary, 2009), so computing the state of a cell $x_i$ at row $N$ requires simulating each step in sequence, without shortcuts (Greenlaw et al., 1995, p. 58). This is not unique to Rule 110: many CA problems are P-complete (Moore, 1997), i.e., not efficiently parallelizable.

Figure 4: A single run of Rule 110. Given the top row, the CA evolves row-by-row according to the 8 rules.

**What does this mean?** Even systems governed by just 8 simple rules can forbid shortcuts.

### 3.2 MANY-BODY MECHANICS

The previous section showed that inherent seriality can arise even in systems governed by a handful of local rules. We now turn to a more realistic setting: Newtonian many-body dynamics.

Consider $N$ particles evolving in $\mathbb{R}^d$ under forces and/or hard collisions. Given initial positions and momenta at $t = 0$, the goal is to predict the particle positions at some later time $t = T$ in a *finite-precision* representation, which reflects practical scientific computation; with unbounded precision, the prediction problem becomes PSPACE-hard. This is the instance depicted in Figure 2(D).

**Many-body mechanics is inherently serial.** Classical mechanics is expressive enough to emulate arbitrary computation. The billiard-ball computer of Fredkin & Toffoli (1982) recreates any Turing machine using nothing but billiard balls elastically reflecting off each other and walls, and Moore (1990) proposes a similar recreation using smooth particle motion in a potential field in $\mathbb{R}^3$. Since the problem of simulating general Turing machines is inherently serial (Greenlaw et al., 1995, p. 58), an algorithm that accurately simulates general physical systems can only be done step by step.

**Seriality in video prediction.** Because videos capture inherently serial dynamics, such as collisions, forecasting the next frame from a sequence of $N$ frames is similarly inherently serial. This task underlies large-scale models for content creation (DeepMind, 2025; Brooks et al., 2024; Wan et al., 2025) and decision making (Bar et al., 2025; Yang et al., 2024).

The computational difficulty in video prediction does not come from the cost of rendering a single next frame. If a model maintains a complete *world state* (e.g., object positions and momenta), stepping that state forward is as efficient as applying a local physical law. The problem arises when

losing track on the state: recomputing it requires replaying the chain of many-body mechanics from the last reliable observation, thus inherently sequential.

In real videos, objects frequently leave the field of view or become occluded, as illustrated in Figure 5. Current large-scale video predictors—typically Transformer-based and trained with next-frame objectives—are optimized only for immediate reconstruction. As a result, they are not encouraged to maintain the state of occluded objects. This can lead to an incomplete world state, effectively rendering the task inherently serial and no longer solvable in a single forward pass by the model. This may help explain why state-of-the-art video generators often produce physically inconsistent results (Kang et al., 2024; Brooks et al., 2024; Wan et al., 2025).

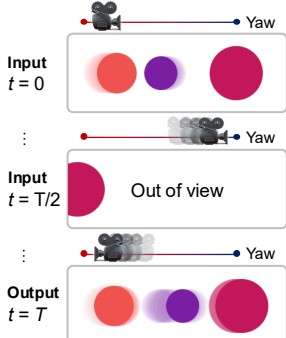

Figure 5: Predicting the frame at time $T$. The intermediate frames may not be observable by camera motion/occlusion.

**What does this mean?** Tasks involving the modeling of system dynamics, including physical simulations and video prediction, likely require serial models to be efficiently solved. For video prediction, the model must maintain the full evolving world state.

## 3.3 P-COMPLETE PROBLEMS

The inherently serial problems given previously have this in common: Given a problem statement of size $\mathcal{O}(n)$, it is immediately obvious how to construct a linear computational graph of length $\mathcal{O}(n)$, such that each node of the graph depends, and only depends, on the previous ones, and each step takes constant time. This matches the intuition of what being "serial" means.

Such problems have been formalized as P-complete problems, of which the most prototypical one is the Circuit Value Problem (CVP) (Ladner, 1975): Given a Boolean circuit with a single output bit, specified as a graph of logic gates and their connections, and an input binary string, what is the output bit? The CVP is robust in that many variations remain P-complete (Greenlaw et al., 1995).

The obvious solution is to perform a topological sort (takes $\mathcal{O}(n)$ time) on the graph of the logic gates, so that each gate depends only on the output values of the previous gates, then evaluate them serially. The CVP embodies the intuition of what an inherently serial problem should be, which is that each step easily depends on the previous steps, but skipping steps is difficult. Since any P problem is efficiently reducible to the CVP, it is P-complete.

This gives us an intuitive rule: If the problem appears to have a linear computational graph similar to the CVP, then it is likely inherently serial. Indeed, this intuition guided us in proving the theorem in Section 3.4. Furthermore, an extensive list of problems is shown to be P-complete (Greenlaw et al., 1995), thus suggesting that inherently serial problems are common in the wild.

## 3.4 SEQUENTIAL DECISION PROBLEMS

The goal in sequential decision problems is to obtain an **optimal policy** $\pi^*(s) = \arg\max_\pi J(\pi)$, where $J(\pi)$ is the expected return under policy $\pi$ in a Markov Decision Process (MDP) with finite horizon $N$, and discount factor $\gamma \in [0, 1]$. Therefore, given a state $s$, the policy outputs an optimal action $a$ from a limited possibilities (discrete action) or with finite precision (continuous action). There are two aspects in this: finding a policy, and executing a policy.

**Executing a policy is inherently serial.** In Appendix G, we construct certain MDPs and prove that in these, any *approximately* optimal policy is inherently serial.

**Finding a policy is inherently serial.** Consider **policy gradient** methods (Sutton et al., 1999; Williams, 1992), including popular variants like PPO (Schulman et al., 2017), widely used in applications such as LLM fine-tuning (Ouyang et al., 2022).

In policy gradient, the parameters of a policy model are improved via gradient descent on return estimates, which must be unbiased for convergence to the optimal policy[3]. Here, we show how an

---

[3]Convergence to suboptimal policies is still possible with biased return estimates (Mu & Klabjan, 2024; Tian et al., 2023).

unbiased return estimation is inherently serial, implying that with parallel estimation the optimal policy is not guaranteed.

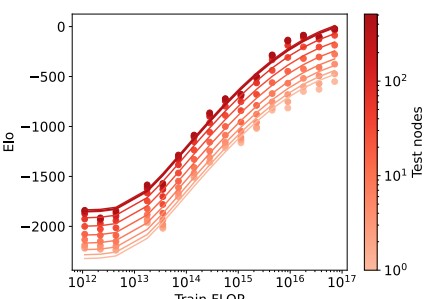

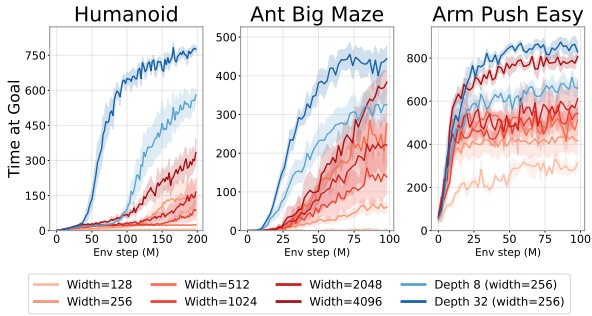

(a) MCTS in Hex board game: Performance improves with more MTCS expansion nodes across all training regimes. Perfect play is only possible with test-time MCTS. Data from Villalobos & Atkinson (2023).

(b) Actor & critic networks' depth vs. width in locomotion & manipulation. A 2× deeper network (8 layers, width 256) outperforms a 16× wider network (4 layers, width 4,096) in Humanoid, a locomotion task with the largest observation & action spaces among the three. Data from Kevin et al. (2025).

Figure 6: Empirical serial scaling on (a) Hex board game and (b) locomotion & manipulation.

If parallel return estimation is possible, one must be able to access state $i$ (and thus reward $r_i$) in fewer than a linear function of $i$ steps. This is unlikely in real-world MDPs, since even simple CA transition rules (Section 3.1) or physical interactions (Section 3.2) prevent such shortcuts[4]. While scaling parallel computation via aggregating multiple trajectories can accelerate convergence by reducing variance (Sutton & Barto, 2018, p. 93), accurate return estimation still requires serial computation.

This inherent seriality motivates model-based reinforcement learning, where returns are computed by unrolling an internal model step by step. For instance, Monte Carlo Tree Search (MCTS)—which increases serial computation via tree expansion and reduces return-estimation bias—has achieved superhuman-level performance in diverse board games (Silver et al., 2016; 2017; Schrittwieser et al., 2020). Villalobos & Atkinson (2023) demonstrate consistent improvements in Hex from increasing MCTS expansion nodes (see Figure 6a). Even in model-free RL, adding serial computation with deeper networks significantly outperform wider (more parallel) ones on locomotion tasks Kevin et al. (2025) (see Figure 6b).

**What does this mean?** Parallel computation cannot substitute for serial computation in RL. Without sufficient serial computation, the optimal policy is not guaranteed.

## 3.5 REASONING QUESTION ANSWERING

Instead of going through an MDP as in RL, math question answering exhibits seriality through step-by-step logical reasoning, irrespective of the answer length, before arriving at the solution. As shown in Figure 2(E), given a question as input tokens, the model autoregressively generates the solution, each step from a limited set of tokens.

**Math QA is likely inherently serial.** Solving grade-school mathematics, GSM8K (Cobbe et al., 2021), which has been used to benchmark reasoning capabilities in LLMs, can be formalized as dependency graphs (Ye et al., 2024). A solution is to traverse the graph in topological order and perform necessary arithmetic operations sequentially (Ye et al., 2024). This resembles the Arithmetic

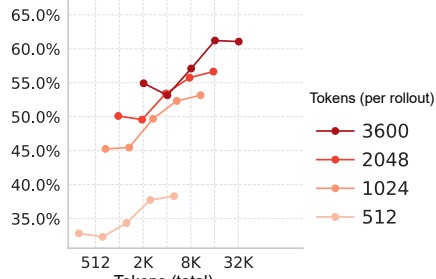

Figure 7: Average benchmark scores over 4 math benchmarks for longer reasoning chains (in different colors) vs. majority voting (as dots along each line). Data from Aggarwal & Welleck (2025).

Circuit Value Problem (Greenlaw et al., 1995, p. 124), a generalization of the standard CVP to arithmetic operations. Like CVP, this problem is P-complete—that is, inherently serial. Since

---

[4]As usual, there is an exponential tradeoff available. One can randomly generate a trajectory and check its consistency with the MDP and policy in parallel, but the expected number of trajectories needed before a consistent one is found grows exponentially with length $n$.

seriality arises even in simple math QA, it likely extends to advanced benchmarks such as AIME and Olympiad-bench, where finding the correct approach is hard.

In **mathematics QA**, as shown in Figure 7, Aggarwal & Welleck (2025) demonstrate that sequential scaling with longer reasoning chains consistently outperforms parallel scaling via majority voting controlled for the same token budget. This pattern holds across mathematical datasets of varying difficulty, including AMC, MATH, AIME, and Olympiad-bench.

Similarly, **science QA** also appears to exhibit inherent seriality. Muennighoff et al. (2025) report that in GPQA Diamond (Rein et al., 2024), a QA benchmark on PhD-level science, sequential scaling yields consistent accuracy improvements, only limited by the model's context window, and is significantly more efficient than parallel scaling (via majority voting), which plateaus.

**What does this mean?** Complex question answering similar to math QA likely requires constructing answers step by step over a computation graph, and thus inherently serial.

## 4 DIFFUSION MODEL'S COMPUTATION IS NOT SERIAL

The Serial Scaling Hypothesis provide a lens to look at machine learning from serial–parallel lens. We have examined serial machine learning problems. We now turn to a diffusion models (Ho et al., 2020; Song et al., 2021; Song & Ermon, 2019), widely used in image/video generation (Brooks et al., 2024) and other vision tasks such as depth estimation (Saxena et al., 2023), and language modeling tasks (Li et al., 2022; Nie et al., 2025; Arriola et al., 2025). While commonly thought as a serial model, we show here for the first time that they are not.

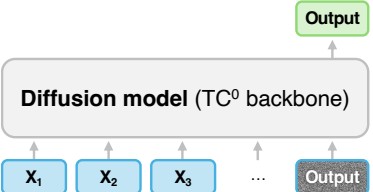

Figure 8: The backbone network takes as input a sequence of tokens, and a single noisy output token. This is repeated until the output token is fully denoised.

Consider a problem with input $x_1, \ldots, x_N$ and fixed output $x_{N+1}$ (as in Figure 2). A diffusion model uses a backbone neural network $\theta$ which takes the inputs and a noisy version of $x_{N+1}$, denoising it over $T$ steps until it becomes the output (see Figure 8). This models the conditional distribution $p_{\text{truth}}(x_{N+1} \mid x_1, \ldots, x_N)$ or concisely $p_{\text{truth}}$.

While the number of denoising steps $T$ is scalable—more denoising steps for finer approximation of $p_{\text{truth}}$, in practice, diffusion models converge rapidly (Ma et al., 2025). For instance, image generation plateaus at 300 steps (Nichol & Dhariwal, 2021), depth estimation shows little difference between 5 and 100 steps (Ravishankar et al., 2024), and language modeling yields similar perplexity for 32 vs. 1024 steps (Austin et al., 2021). With distillation, 1–4 steps suffice without much loss (Yin et al., 2024; Liu et al., 2024; Song et al., 2023; Lin et al., 2024; Salimans & Ho, 2022). Such rapid convergence suggests that the effective computation depth of diffusion models is low. It would be surprising if the underlying computation were truly serial.

**Diffusion models with a $\mathsf{TC}^0$ backbone can only solve problems in $\mathsf{TC}^0$.** Previous work showed that a backward diffusion process converges to $p_0$ at the rate $TV = \mathcal{O}(d/T)$ (Li & Yan, 2024, Thm. 1), where $TV$ is the total variation between $p_0$ and the denoising $p_{\theta,0}$, and $d$ is the intrinsic dimension of $x_{N+1}$. Building on top of it, we obtained the following theorem, the formal statement and proof of which are in Appendix F:

**Theorem 4.1** (Informal). *If a problem can be solved by a diffusion model with a $\mathsf{TC}^0$ backbone with high probability with infinite diffusion steps, then the problem itself is in the parallelizable class $\mathsf{TC}^0$.*

**What does this mean?** Diffusion models only provide a *constant* amount of additional serial computation. The above theorem precludes the use of diffusion models as a scalable means of increasing serial computation. Unlike CoT, which genuinely adds serial compute (see Appendix E), diffusion models do not. This may explain the empirical mediocre performance of diffusion language modeling as output length increases (Austin et al., 2021; Lou et al., 2024; Gulrajani & Hashimoto, 2023; Sahoo et al., 2024).

**Limitation.** The theorem doesn't apply: (1) If solution space grows in intrinsic dimension. (2) If the backbone is either poorly trained or trained under a different objective. This leaves open the possibility of serial-compute-scalable generative models beyond score-based diffusion.

## 5 ACKNOWLEDGEMENTS

We thank Alexei Efros, whose critiques of deep learning theory helped ground us and motivated us to make this paper friendlier to non-theoreticians. We are grateful to William Merrill and Anant Sahai for their invaluable technical feedback and corrections that significantly strengthened our theoretical analysis. We also thank Angjoo Kanazawa, Justin Kerr, Amil Dravid, Yossi Gandelsman, and Yuatyong Chaichana for their thoughtful comments and suggestions that helped improve the clarity and presentation of this work. This work is partially supported by the ONR MURI N00014-21-1-2801. YL was supported by a gift to the Center for Human-Compatible AI at Berkeley from Coefficient Giving (formerly Open Philanthropy) during the development of this work.

## 6 REPRODUCIBILITY

No new data or code has been used to support this paper. Some new theorems are introduced in this paper, and their proofs are in the appendix.

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

## A    LIMITATIONS

Our conclusions rely on the widely-held but unproven assumption that TC $\subsetneq$ P (Assumption 2.3). If this assumption is disproven, our formalization of the serial/parallel dichotomy would be invalid. Moreover, our theoretical arguments apply only to the general case; although the intuition may extend to practical settings, such generalization is not guaranteed. In particular, the average-case complexity (Bogdanov et al., 2006) of many problems could differ or even allow parallel solutions.

For especially hard problems (e.g., NP \ P and beyond), exponential cost—rather than serial depth—may dominate as the primary bottleneck. Thus, our focus is on real-world problems of practical difficulty.

Our theorem on diffusion models applies only when the output dimension remains fixed. If it grows with problem size, the result may not hold—though current empirical evidence in language modeling does not suggest strong serial scaling. The theorem also assumes a well-trained backbone using a score-matching objective, and may not apply to poorly trained models, especially early in training.

Finally, beyond theoretical and circumstantial evidence, more empirical work is needed to quantify the degree of seriality in practice—particularly the benefits of increased serial compute. A promising direction is to benchmark real-world tasks under varying ratios of serial and parallel computation.

## B    A BRIEF HISTORY OF SCALING

The success of modern machine learning has been driven by scaling—bigger models, more data, and especially more parallel compute. Hardware has shifted from CPUs to massively parallel GPUs; architectures have moved from RNNs to highly parallelizable Transformers; and algorithms increasingly exploit parallelism for efficiency.

|  | Serial | Parallel |
|---|---|---|
| **Scaling** | Depth | Width |
| **Architecture** | RNN | Transformer |
| **Algorithm** | RL | Imitation learn. |
| **Hardware** | CPU | GPU |

Table 2: Examples of serial and parallel approaches.

However, this focus on parallel scaling has a blind spot: it assumes that parallel and serial computation are interchangeable. In reality, for many problems, only increasing serial compute—allowing models to perform more sequential steps—yields further progress. This is especially true for tasks with inherent temporal or causal dependencies. As we show in this paper, recognizing the limits of parallel scaling and the necessity of serial computation is crucial for the next phase of progress in machine learning.

## C    POTENTIAL MISCONCEPTIONS

### C.1    ARCHITECTURE VS. INFERENCE METHODS

One may be confused by the statement that "Transformers are not serial," followed by "Chain of Thought is serial." This appears to be a contradiction. It is not, though it may appear so due to our omission of words. The first statement really should be "If you fix a Transformer's parameters and give it problems of size $\mathcal{O}(n)$, but only run it for $\mathcal{O}(1)$ forward passes, then you cannot solve inherently serial problem-families.". Similarly, the second statement really should be "If you fix a Transformer's parameters and give it problems of size $\mathcal{O}(n)$, and also run it for $\mathcal{O}(n)$ forward passes, then you can solve inherently serial problem-families.".

This is a special case of a general phenomenon: people confuse architecture and inference methods, meshing them all together like two lumps of clay gently melding under the desert sun. This is unfortunate but understandable because, in practice, architecture and inference couple together strongly, and this confusion is also encouraged by things like "architecture–deployment co-design". The fact is that the abstractions leak. How you use a model for inference depends on its architecture. How you design a new model architecture depends on what you expect its inference method to be. This is a good thing, but it may cause people to confuse architecture and inference.

Fundamentally, we consider an "architecture" to be a **class** that can have **instances**. For example, `gpt2` is an instance of the class `CausalTransformer`. Each class-instance should be imagined as an "atom of computation". An atom has mass and volume. An atom can be copied and put

into many molecules. What an atom cannot do is be divided or changed. Similarly, an instance of the "Transformer architecture" is an "atomic" Transformer. This atomic Transformer does have "subatomic structures", such as its individual layers. However, since we are not performing complicated model surgeries, such as taking off its head, reading out the tensor from its middle layers, etc., we can ignore all of its subatomic structures and assume it is simply atomic, perfectly indivisible, unchangeable, eternal. If I give you an atomic Transformer, and you want to do something with it, you *must* perform exactly one single forward pass, because you cannot stop half-way (since you don't have access to its subatomic structures). This has a fixed seriality. It may be 48 layers or 96 layers, and you can pick bigger and bigger atoms from the same class with no upper bound. But for a *given* particular instance from the class, the seriality is $\mathcal{O}(1)$, and cannot possibly grow like $\mathcal{O}(n)$.

Still, even though the atomic Transformer is unchanging, it can be used in many molecules, and the molecules can change. It can be used in a Chain of Thought molecule, and this molecule would grow like a polymer, taking on a length of $\mathcal{O}(n)$ in reaction to a length $\mathcal{O}(n)$ input.

In short, a single architecture may be used on its own in a fixed number of forward passes, or a number of forward passes growing linearly with the problem size, or in a tree search, a beam search, etc. Inference methods are separate from model architecture. In this paper, we discuss both architectures and inference methods. We have found that Transformers, SSMs, and diffusion models are all not serial when the inference method is just a single forward pass, but can become serial when the inference method allows for more serial compute, such as Chain of Thought inference.

## C.2 SSM IS NOT RNN

We class SSMs as *not* inherently serial, while RNNs as inherently serial. This is not a mistake, and in fact, a very valuable illustration of how an architecture can have an illusion of seriality.

First, in what sense is RNN serial, while a Transformer is not serial? Recall the title of the original paper on Transformers: "Attention is all you need" (Vaswani et al., 2017). What did they mean by "all you need"? What was removed? Back in 2017, the standard sequence transduction model was a pair of RNNs connected in the middle by the attention mechanism. The encoder RNN processes the input sequence one token at a time, then the decoder RNN produces the output sequence, again one token at a time. The key problem is that, while both an RNN and a Transformer must decode tokens one-by-one, an RNN must encode tokens *also* one-by-one, whereas a Transformer can encode tokens all-in-one-go. The animating spirit of the Transformer architecture was to remove everything that stands in the way of all-in-one-go, but preserve everything that allows RNNs to work so well.

So they removed recurrence and preserved attention. Their title meant "attention is all you need (and recurrence is not)". This is exactly what our paper is arguing against: attention is not all you need, and inherently serial processing is necessary. It can be done in many ways: recurrence in architecture, CoT in inference strategy, or some other method, but it cannot be avoided.

So now we come to SSM. SSM is not RNN because it is not recurrent in an inherently serial way. It was designed with the same spirit as the Transformer and suffers the same issue.

Concretely, consider the problem of ingesting an input sequence: $x_1, x_2, \ldots, x_n$, and the task is to output the next token $x_{n+1}$. For a Transformer with 96 layers to accomplish the task, it does so by 96 sequential steps. Each step is very wide but not deep. In contrast, an RNN with 96 layers must process $x_1$ first, then $x_2$, and so on. It requires $96n$ sequential steps that cannot be done in parallel. Suppose one attempts to do them "in parallel", then there is an immediate problem: To run the RNN on $x_4$, one needs the internal states of the RNN just after it has processed $x_3$, and to do that, it needs to have processed $x_2$, etc. Abstractly, the operation of an RNN looks like:

$$f_\theta(x_n, f_\theta(x_{n-1}, \ldots, f_\theta(x_2, f_\theta(x_1, s_0)))),$$

where $s_0$ is the initial internal state.

In short, because the internal states of an RNN change nonlinearly and unpredictably, one cannot skip the steps. This allows RNN to solve inherently serial problems without chain of thought, but at the price of taking wall-clock time $\mathcal{O}(n)$ to even output its first token.

The animating idea of SSM is precisely to remove the inherent seriality of RNN. The idea is that if the internal states change linearly and predictably, then one *can* skip the steps. Concretely, the

operation of an SSM is of the form:

$$\text{output}_k = f_\theta(M^{k-1}x_1 + M^{k-2}x_2 + \cdots + M^1 x_{k-1}, x_k)$$

since its internal states change linearly in a data-independent way. This means that to run an SSM on input $x_4$, one does not need to know the internal state of the SSM after processing $x_3$. This allows it to be just as parallel as a Transformer, and thus just as lacking in seriality.

### C.3 TRAINING VS. INFERENCE

Throughout most of the paper, the main contrast is the parallel vs. serial computation contrast. We do not mean this to be a contrast between the parallel phase of training vs. the autoregressive phase of inference, which is how modern GPT-like transformers are produced and used.

Concretely, consider initializing a language model with 96 layers. The training algorithm samples a chunk of text (32,768 tokens long) from the training corpus. The model then performs a single forward inference simultaneously on all tokens, such that it only takes 96 steps of time—one step per layer. Now, during inference, the model must then generate one token at a time, so the same 32,768 tokens would take 3,145,728 steps of time, which is a vast expansion.

Yet, this is not the main focus of the paper. The paper mostly talks about the theoretical and empirical results concerning *learned and frozen* models. The paper does talk about training, but only occasionally and informally, by sketching out intuitive analogies with inference, without rigorous theoretical justification.

This is not due to an intention to mislead. Indeed, we recognize that there is a parallel training vs. autoregressive inference contrast, and we have attempted to get some theoretical handle on this contrast. We failed. Because learning theory is extremely difficult, we could neither find theorems proven by people that came before us, nor prove theorems ourselves. Therefore, we stayed within inference. Fortunately, the parallel vs. serial contrast is already sharp even within inference, with both clear theoretical and empirical results, allowing us to write the paper.

We believe strongly that in the future, the parallel vs. serial contrast will also be shown for training, and leave this as work for the near-posterity.

## D $\mathsf{TC}^0$ AND $\mathsf{TC}$ CLASSES

**Definition D.1.** Let $i \in \mathbb{N}$. A decision problem $L \subseteq \{0,1\}^*$ is in the complexity class $\mathsf{TC}^i$ if there exists a family $\{C_n\}_{n \in \mathbb{N}}$ of Boolean circuits such that:

- Each circuit $C_n$ decides whether $x \in L$ for all $x \in \{0,1\}^n$.

- Each circuit $C_n$ has size polynomial in $n$, i.e., $|C_n| = \mathcal{O}(n^k)$ for some constant $k$.

- Each circuit has depth $\mathcal{O}(\log^i n)$.

- The circuit gates are of unbounded fan-in AND, OR, NOT, and MAJORITY gates. A majority gate outputs 1 if more than half of its inputs are 1.

- The family $\{C_n\}$ is L-uniform, where L stands for "logspace". This means that there exists a deterministic Turing machine that, on input $(\underbrace{11\cdots1}_{n \text{ repeats}}, i)$, outputs the $i$th bit of the description of $C_n$ using a working tape of length only $\mathcal{O}(\log n)$.

In addition, a decision problem $L \subseteq \{0,1\}^*$ is in the complexity class $\mathsf{TC}$ if it is in the union of all classes $\mathsf{TC}^i$ over $i \in \mathbb{N}$, that is,

$$\mathsf{TC} = \bigcup_{i \in \mathbb{N}} \mathsf{TC}^i. \tag{1}$$

In the literature, $\mathsf{TC}$ is also called $\mathsf{NC}$, or "Nick's Class."

Define the L-uniform many-one reduction relation $\leq_m^{\mathsf{L}}$ as follows: Given two languages $L, L'$, that is, two sets of binary strings, we say $L \leq_m^{\mathsf{L}} L'$ if and only if there exists a function $f$ such that $x \in L$ if

and only if $f(x) \in L'$, and the function $f$ is computable by a deterministic Turing machine with a logspace working tape.

A decision problem is P-complete if and only if there exists a Turing machine that decides it in polynomial time, and any problem decidable in polynomial time can be L-uniformly many-one reduced to it.

# E  SERIAL CAPABILITIES OF MODERN MACHINE LEARNING METHODS

Empirically, we see a tradeoff in modern machine learning architectures. Some have highly paralleliz-able computation graphs, such as MLPs, Transformers, and SSMs, but do not solve inherently serial problems. Others can solve inherently serial problems, but cannot be parallelized, such as RNNs, repeating layers, and CoT.

In terms of computational complexity, MLPs have been formalized as a family of Boolean circuits with threshold gates called $\mathsf{TC}^0$ (Parberry & Schnitger, 1988). Consider a family of MLPs with constant $\mathcal{O}(1)$ depth, $\mathrm{poly}(n)$ neurons per layer, and $\mathcal{O}(1)$ numerical precision. A problem is in $\mathsf{TC}^0$ if and only if it is decidable by one forward pass through an MLP under this setup.

It is widely suspected that $\mathsf{TC}^1$ is strictly larger than $\mathsf{TC}^0$. Therefore, any $\mathsf{TC}^1$-hard problem requires computations deepening at a rate of at least $\mathcal{O}(\log n)$. We may consider the problem as being serial in a weaker, logarithmic sense. A simple example of $\mathsf{TC}^1$ is the **word problem** of the symmetric group on 5 elements $S_5$: Given $g_1, g_2, \ldots, g_n \in S_5$, find $\prod_i g_i$ (Liu et al., 2022). Intuitively, this is because $S_5$ is not a solvable group, and therefore there is no better way to multiply its elements than via a binary tree, which has $\mathcal{O}(\log n)$ layers. Attempting to perform this with just $\mathcal{O}(1)$ layers would effectively require one to memorize the entire $S_5^n \to S_n$ multiplication table, which scales exponentially as $e^{n \log(120)}$, an exponential depth-width trade-off.

To solve the word problem of $S_5$ by a Transformer in one forward pass, we present it with input tokens $g_1, g_2, \ldots, g_n, y$, where $y$ is a special token. The Transformer's output at $y$ is read out as the answer. Similarly, we may solve such a problem by presenting the same input tokens to a State Space Model (SSM) or a Recurrent Neural Network (RNN). Since a Transformer with $\mathcal{O}(1)$ layers and $\mathrm{poly}(n)$ dimensions can only solve problems in $\mathsf{TC}^0$ (Merrill & Sabharwal, 2023b) in one forward pass, it cannot solve the word problem of $S_5$ that lies outside $\mathsf{TC}^0$.

On the contrary, for an RNN, we can write the multiplication table of $S_5^2 \to S_5$ directly into its weights, so it can solve the problem by unrolling for $\mathcal{O}(n)$ recurrence steps with $\mathcal{O}(1)$ layers and $\mathcal{O}(1)$ dimensions. Intuitively, the hidden states of an RNN keep track of the progress of multiplication as it performs the forward passes. However, RNN's recurrence state dependency renders it non-parallelizable. Several families of SSMs were developed as a compromise that still have recurrence on hidden states, like an RNN, while making forward passes parallelizable, like a Transformer. The prototypical SSM architecture is Mamba (Gu & Dao, 2023), though there are many others.

Unfortunately, it has been proven that there is not yet a "free serial-compute lunch," in the sense that the main families of SSMs proposed so far still cannot solve the word problem of $S_5$ under the constraints of $\mathcal{O}(1)$ layers, $\mathrm{poly}(n)$ dimensions per layer, and one forward pass. Despite its apparent recurrence, the hidden state offered by an SSM is weaker in a computational sense than that offered by an RNN (Strobl et al., 2023; Merrill & Sabharwal, 2023a; Merrill et al., 2024). This theoretical fact rhymes with the empirical fact that in practice, MLPs, Transformers, and SSMs are all more parallelizable than RNNs.

Only a genuinely serial method has been shown to go beyond the $\mathsf{TC}^0$ class. In addition to RNNs, this includes repeating layers and CoT. Repeating layers for $\mathcal{O}(\log n)$ times enables a standard Transformer to solve tasks in the $\mathsf{TC}^1$ class (Merrill & Sabharwal, 2025). With $\mathrm{poly}(n)$ CoT, requiring multiple forward passes before producing a final answer, a Transformer can be lifted from $\mathsf{TC}^0$ to P (Feng et al., 2023; Li et al., 2024). As discussed in Section 3.5, the power of CoT has been well-attested in practice by the improved performance of reasoning models in complex math and science tasks.

Such a uniformity restriction is necessary for technical reasons. Specifically, it is necessary because one may hide a large amount of computation into a small circuit that requires a long time to find. The final circuit produced might run in time $\mathcal{O}(\log n)$, but if the time required to find such a circuit requires time $2^{\mathcal{O}(n)}$, then this would not be parallel—in the sense used throughout this paper.

# F    DIFFUSION IS IN $\mathsf{TC}^0$

In this section, we consider the non-uniform $\mathsf{TC}^0$ class, in contrast to the usual uniform classes. Note that uniform $\mathsf{TC}^i \subseteq$ non-uniform $\mathsf{TC}^i$. We prove that many diffusion models are restricted within that class. We need to assume non-uniformity because, at a certain point in the proof of the main theorem, we merely prove that something exists, without showing that it is also efficiently computable. We will highlight this in the proof.

## F.1    PROBLEM SETTINGS

An abstract language is simply a set of sentences made of letters. Formally:

- An **alphabet** $\Sigma$ is a finite nonempty set. Each element in the alphabet may be called a **letter** or a **token**.
- A **sentence** in an alphabet $\Sigma$ is a finite sequence of elements of $\Sigma$.
- A **language** in an alphabet $\Sigma$ is a set of sentences in the alphabet $\Sigma$.

An abstract language is more than a natural language. A sentence in a natural language is a sequence of tokens. An image, divided into patches and tokenized, becomes a sentence in an abstract language. A video, divided into frames and patches and tokenized, becomes a sentence in an abstract language.

We define the **deterministic prefix language modeling problem**: given a sequence of tokens $x_1, \ldots, x_n$, compute the next token $x_{n+1}$. This is a deterministic formalization of next-token prediction, the dominant paradigm in language modeling since the GPT-2 of 2019. This can also be cast into a decision problem: given input of size $n$, $x_1, \ldots, x_n$, decide whether $x_{n+1}$ is the correct continuation.

Most language model benchmarks can be cast into this form, at least when without chain of thought. For example, consider a problem from the GSM8K benchmark "Tina buys three 12-packs of soda for a party ... How many sodas are left over when the party is over?" has only one correct continuation, "11".

The definition has some issues. Some problems may have more than one correct answer. Some answers may occupy more than one token. We may allow chain of thought. Language modeling is not restricted to prefix modeling. A touted benefit of diffusion language modeling is that, unlike GPT models, it can generate any number of tokens anywhere in a sequence conditional on other parts of the sequence.

Define **nondeterministic masked sequence modeling problem** as follows. Consider a sequence of tokens $x_1, \ldots, x_n$, some of which are masked. The task is to compute a sequence of tokens that can acceptably fill in the masks.

For example, a game of 24 may be cast into a nondeterministic masked modeling problem[5] as follows: "1, 3, 9, 9, [M][M][M][M][M][M][M] = 24". This has more than one acceptable answer, such as "9 3 ÷ 11 × 9 -".

As another example, solving a problem with chain of thought is unmasking an entire block of masked tokens, where any unmasking is acceptable as long as the tokens between "Final answer: ... [END]" are correct.

Given a problem instance, we can train a model such that, conditional on the problem instance, it would sample one point from possibly many points that form a solution manifold, from which the answer can be recovered by discretizing.

---

[5]Brackets would create different lengths for the masked sequence. We avoid them by reverse Polish notation.

We need to make the discretization step as simple as possible, to avoid secretly performing part of the problem-solving in the discretization step. Consequently, we will only consider cases where the masking pattern is fixed, and the discretization is performed using a fixed amount of compute.

As a concrete illustration, suppose that we use a diffusion video model to evaluate an arithmetic expression. The input is a sequence of frames presenting the arithmetic expression to be evaluated, and the masked tokens are the frames illustrating an animated process that draws out the answer. The solution might be written in many variable ways, creating the solution manifold. The discretization step can be done by taking the pixel-wise nearest neighbor of the last frame's digits.

Our overall framework for sequence modeling is:

$$\text{true answer} \xleftarrow{\text{discretization}} \text{solution manifold} \xleftarrow{\text{stochastic sampling}}$$

We will show that if the dimension of the solution manifold does not grow quickly, and the stochastic sampling is sufficiently parallelizable and accurate, then we can derandomize it to obtain a parallelizable deterministic algorithm for obtaining the true answer, thus showing that the original problem is parallelizable, not serial.

## F.2 DIFFUSION MODELING

There are several equivalent formulations of diffusion modeling. We use the score-matching formulation. In this formulation, given a probability density $\rho_0$ to be modeled, and a **noise schedule** $\sigma_1, \sigma_2, \ldots, \sigma_T$, we define the forward diffusion process by adding an increasing amount of Gaussian noise to $\rho_0$:

$$x_0 \sim \rho_0, \quad x_t | x_0 \sim \mathcal{N}\left(\sqrt{1 - \sigma_t^2} x_0, \sigma_t^2 I\right)$$

Let $\rho_t$ be the probability density of $x_t$. Any (true) **score function** at time $t$ is $f^*(t, x) := \nabla \log \rho_t(x)$. To score-match is to find an (approximate) score function $f_\theta$, such that $f_\theta(t, x) \approx f^*(t, x)$. In the sense that the **average score modeling error** is low:

$$\epsilon_{\text{score}} := \sqrt{\frac{1}{T} \sum_{t=1}^{T} \mathbb{E}_{x_t \sim \rho_t} \left[ \| f_\theta(t, x_t) - f^\star(t, x_t) \|_2^2 \right]}$$

Given a score approximator $f_\theta$ and a noise schedule $\sigma_t$, the **Score-Matching with Langevin Dynamics (SMLD)** (Song & Ermon, 2019) sampler performs a backward diffusion accordingly, sampling a sequence of $\hat{x}_T, \hat{x}_{T-1}, \ldots, \hat{x}_0$. Let $\rho_{SMLD,t}$ be the probability density of $\hat{x}_t$. The theoretical basis for diffusion modeling is that, at the limit of continuous noise schedule – infinitely many steps, each adding an infinitesimal amount of noise – and the limit of perfect score function, $\rho_{SMLD,0}$ would converge to be exactly equal to $\rho_0$.

Generally, the difference between $\rho_{SMLD,0}$ and $\rho_0$ can be understood as consisting of two parts: a discretization error, caused by taking finitely many small steps, instead of infinitely many infinitesimal steps; a score-matching error part, caused by $f_\theta \neq f^*$.

Intuitively, the discretization error converges to zero as $T \to \infty$, while the score-matching error grows with $\epsilon_{\text{score}}$ and $T$. In the next section, we quote a theorem from the literature that makes this intuition precise.

## F.3 THE FUNDAMENTAL THEOREM (LI & YAN, 2024, THM. 2)

Intuitively, a point is easier to model than a line, which is easier than a plane, and so on. This can be made precise by the intrinsic dimension of the support of a probability distribution to be modeled.

**Definition.** The **support** of a probability distribution $\rho_0$ is $\text{supp}(\rho_0)$. It is the smallest set that satisfies $\Pr_{x \sim \rho_0}(x \in \text{supp}(\rho_0)) = 1$.

Let $T$ denote the number of SMLD denoising steps. It is a positive integer.

**Definition.** Given a base space $\Omega$, and two probability densities $p, q$ over it, their **total variation** is

$$TV(p, q) := \int_\Omega |p(x) - q(x)| dx$$

**Definition.** Let $X$ be an arbitrary subset of $\mathbb{R}^n$. Let $\epsilon > 0$. The **discrete intrinsic dimension** is

$$d_\epsilon(X) := \frac{\log N_\epsilon(X)}{\log \frac{1}{\epsilon}},$$

where $N_\epsilon(X)$ is the minimal number of radius-$\epsilon$ balls necessary to entirely cover $X$. As $\epsilon \to 0$, if the discrete intrinsic dimension converges, then what it converges to is the **intrinsic dimension**.

What does it mean?

Consider a 2-dimensional square $X$ in $\mathbb{R}^n$. As we repeatedly halve the value of $\epsilon$, each halving would require 4 times as many radius-$\epsilon$ balls to cover the same square. Thus, $d(X) = \frac{\log 4}{\log \frac{1}{1/2}} = 2$. This is true no matter if $n = 3$ or $n = 300$, and no matter what side length the square has. Thus, the intrinsic dimension recaptures our intuition. However, it is well-defined for more than smooth manifolds. It is well-defined for more general sets, such as fractals. The expression for intrinsic dimension is deeply related to metric entropy, and has applications in statistics and information theory. (Wainwright, 2019, Chap. 5)

However, the intrinsic dimension is not accessible in practice. For example, consider a line segment $X$ in $\mathbb{R}^{100}$. It has $d_\epsilon(X) \to 1$ as $\epsilon$ decreases. However, when $\epsilon$ gets small enough, suddenly $d_\epsilon(X)$ starts converging to 2 instead. What happened? It turns out the line segment is not really a line segment. At a high enough zooming level, it is actually a cylindrical tube. However, when $\epsilon$ got even smaller, suddenly $d_\epsilon(X)$ starts converging to 1 again. It turns out the cylindrical tube is actually a tightly-wound helix curve.

What is accessible in practice is the discrete intrinsic dimension $d_\epsilon(X)$. Intuitively, it is intrinsic dimension $X$ if we are not allowed to look more closely than $\epsilon$. Another way to intuit it is by taking $X$ and constructing a low-dimensional $\epsilon$-skeleton $X_\epsilon$ of it. Any point in $X$ is within distance $\epsilon$ of $X_\epsilon$. The intrinsic dimension of the "dehydrated skeleton" of $X$ is approximately $d_\epsilon(X)$.

**Theorem F.1.** *There exists constants $c_M, c_\epsilon, c > 0$, such that the following is true. Fix any positive integer $T > 0$. Fix $\epsilon = T^{-c_\epsilon}$. Let $\rho_0$ be the target distribution. If the target distribution has first order moment that is bounded by a polynomial in $T$:*

$$\mathbb{E}_{x_0 \sim \rho_0} [\|x_0\|_2] \le T^{c_M}.$$

*and we have a score network $f_\theta(x, t)$ such that it achieves average score modeling error $\epsilon_{score}^2$, then there exists a SMLD sampler schedule that takes $T$ steps, such that the result of SMLD sampling has distribution $\rho_{SMLD,0}$, and*

$$TV(\rho_0, \rho_{SMLD,0}) \le c \left( \frac{d_\epsilon(\mathrm{supp}(\rho_0))}{T} (\log T)^3 + \epsilon_{score} \sqrt{\log T} \right)$$

The two terms correspond to error caused by taking discrete steps, and error caused by drifting away from score matching error.

## F.4 MAIN THEOREM

We formalize a model of non-uniform $\mathsf{TC}^0$ diffusion modeling.

Given a task, we define what it means to solve it by nondeterministic masked sequence modeling. Each task instance is specified by a sequence of $n$ tokens, written as $x_1, \ldots, x_n$. Of these, $k_n$ tokens are fixed inputs to the model to specify the problem, and $n - k_n$ tokens are masked, which the model would denoise. The denoised tokens are discretized to a final answer.

To avoid surreptitiously hinting the diffusion model the answer, the masked tokens are always in the same place, which, WLOG, we assume comes at the end. Similarly, we require that for each $n$, $k_n$ is fixed. Finally, we require the discretization step to require constant compute, to avoid secretly offloading problem-solving computation from the diffusion model to the discretization algorithm.

Given a task, a $\mathsf{TC}^0$ family of score-networks $f_{\theta,n}$ for such a task is a sequence of networks $f_{\theta,1}, f_{\theta,2}, \ldots$, such that:

- Each $f_{\theta,n}$ takes as input $n+1$ elements $x_1, \ldots, x_n, t$, and produces an output $f_{\theta,n}(x_{k_n+1}, x_{k_n+2}, \ldots, x_n | t, x_1, \ldots, x_{k_n})$.
- The family $f_{\theta,n}$ has $\mathcal{O}(1)$ depth, $\mathrm{poly}(n)$ width, and $\mathcal{O}(1)$ precision.

**Comment.** The holds for any family of score-networks for which a single forward pass is in $\mathsf{TC}^0$. This includes, for example, Transformers and state-space models (Merrill & Sabharwal, 2023b; Merrill et al., 2024).

For notational convenience, we will thenceforth assume that in our task, there is one true solution per problem. If there are multiple true solutions per problem, then our proof shows that the task of "finding at least one true solution for the original task" is $\mathsf{TC}^0$.

Since a diffusion model may solve a problem only with high probability, instead of solving it deterministically, we make the following definition:

A task is **solved with constant probability bound** if there exists some $\delta > 0$, such that for each input token sequence $x_1, \ldots, x_n$, let $x_{\text{correct}}$ be the correct discrete solution, then

$$p(x_{\text{correct}} | x_1, \ldots, x_n) > p(x' | x_1, \ldots, x_n) + \delta, \quad \forall x' \neq x_{\text{correct}}. \tag{2}$$

As noted before, there are 2 places where a diffusion model can incur error in its solution. The first place is due to using discrete time-steps during the backward diffusion process. The second place is due to score-matching error compared to the true forward diffusion process from the solution manifold.

We are ready to state the theorem. We apologize for the amount of complicated epsilon-delta formality in the statement, but it is what it takes to make it rigorous.

**Theorem F.2.** *Given a task, and a $\mathsf{TC}^0$ family of score-networks $f_{\theta,0}, f_{\theta,1}, \ldots$, for solving the task, by SMLD with $f_{\theta,n}(x_{k_n+1}, x_{k_n+2}, \ldots, x_n | t, x_1, \ldots, x_{k_n})$.*

*Let the $n$-th solution manifold of the task be $X_n$. Assume:*

1. *There exists a constant integer $T > 0$, a constant real number $\epsilon_{score} > 0$, a small constant real number $\delta > 0$, and a sequence of probability distributions $\rho_{0,n}$ on $X_n$, such that*

2. *Letting $c, c_\epsilon, c_M$ be the constants used in the statement of Theorem F.1, $\epsilon = T^{-c_\epsilon}$, and $\epsilon_{score,n}$ be the average score modeling error of using $f_{\theta,n}$ to score-match the forward diffusion process on $\rho_{0,n}$,*

3. *For all $n = 1, 2, 3, \ldots$, we have*

$$\mathbb{E}_{x_0 \sim \rho_{0,n}}[\|x_0\|_2] \leq T^{c_M},$$

$$\epsilon_{score,n} \leq \epsilon_{score},$$

$$c \left( \frac{d_\epsilon(X_n)}{T} (\log T)^3 + \epsilon_{score} \sqrt{\log T} \right) \leq \frac{1}{2} - \delta.$$

*Then the task is in the non-uniform $\mathsf{TC}^0$ class.*

*More generally, if the family of score-networks is a $\mathsf{TC}^k$ family, and assume:*

1. *There exists a non-negative integer $k$, a sequence $T_1, T_2, \ldots$ such that $T_n = \mathcal{O}((\log n)^k)$, a sequence $\delta_n$ such that $1/\delta_n = \mathrm{poly}(n)$, such that*

2. *Letting $c, c_\epsilon, c_M$ be the constants used in the statement of Theorem F.1, $\epsilon_n = T_n^{-c_\epsilon}$, and $\epsilon_{score,n}$ be the average score modeling error of using $f_{\theta,n}$ to score-match the forward diffusion process on $\rho_{0,n}$,*

3. *For all $n = 1, 2, 3, \ldots$, we have*

$$\mathbb{E}_{x_0 \sim \rho_{0,n}}[\|x_0\|_2] \leq T_n^{c_M},$$

$$c\left(\frac{d_{\epsilon_n}(X_n)}{T_n}(\log T_n)^3 + \epsilon_{score,n}\sqrt{\log T_n}\right) \leq \frac{1}{2} - \delta_n.$$

*then the task is in the non-uniform $\mathsf{TC}^k$ class.*

*Proof.* The proof of the case for $\mathsf{TC}^k$ is essentially the same as the case for $\mathsf{TC}^0$, with more cumbersome notations. Thus, we only prove the case for $\mathsf{TC}^0$ explicitly.

Using the big list of assumptions, we can apply Theorem F.1 directly, and conclude that, if we use SMLD with $f_{\theta,n}$ as the score network, for $T$ steps, we would sample from a distribution $\rho_{SMLD,0,n}$ that satisfies

$$TV(\rho_{0,n}, \rho_{SMLD,0,n}) \leq 1/2 - \delta$$

In particular, this means that after discretization, we obtain the correct solution with probability at least $1/2 + \delta$. This then implies that the task is solved with constant probability bound. Now we can derandomize this family, obtaining a $\mathsf{TC}^0$ family of Boolean circuits that solves the problem deterministically. The details of the derandomization method appear in (Hajnal et al., 1993, Prp. 4.2). It goes as follows:

for each length $n$, we replicate the network $k(n)$ times. Each network must take one seed. For each choice of $k(n)$ seeds $s_1, \ldots, s_{k(n)}$, we have a particular deterministic model:

$$x \mapsto \mathrm{majority}(f(x, s_1), f(x, s_2), \ldots, f(x, s_{k(n)})).$$

If $s$ is randomly sampled, then let the probability that $f(x, s)$ is wrong be upper-bounded by a constant $p < 1/2$. By Hoeffding's inequality (Hoeffding, 1963), the probability that the majority vote is correct is $\geq 1 - e^{-2k(n)(p-1/2)^2}$.

Sample $k(n)$ random seeds, and fix them. This provides a deterministic model. Now, we try this deterministic model on every single possible input of length $n$. There are only $\#\mathrm{vocab}^n$ of them. If we set $k(n) \geq \frac{\log(\#\mathrm{vocab})}{2(p-1/2)^2}$, then by the union bound, the probability that the majority vote is correct on *all* inputs of length $n$ is nonzero. Thus, there exists a specific choice of random seeds $(s_1, \ldots, s_{k(n)})$ that makes the compound model correct on all inputs of length $n$.

This construction is nonuniform precisely in the part where we have only shown the choice of seeds[6] exists. To actually find these seeds may take exponential time. □

## F.5 INTERPRETATION

At the high level, to satisfy the assumptions, one needs to have a good enough approximate score with an error $\epsilon_{\mathrm{score}}$. Then, choose $T$ (or $T_n$) such that

$$c\left(\frac{d_\epsilon(X_n)}{T}(\log T)^3 + \epsilon_{\mathrm{score},n}\sqrt{\log T}\right) \leq 1/2 - \delta$$

For constant $d_\epsilon(X_n)$, we have the following interpretations:

**Perfect diffusion model is non-uniform $\mathsf{TC}^0$.** Consider the case if we have a perfect score function $\epsilon_{\mathrm{score}} = 0$, we have $c\left(\frac{d_\epsilon(X_n)}{T}(\log T)^3\right) \leq 1/2 - \delta$. We can select $T = O(d_\epsilon(X_n)) = O(1)$, hence, the task belongs to the non-uniform $\mathsf{TC}^0$ class. This is the same finding as Liu (2025).

---

[6] In complexity theory, this choice of seeds is an "advice string".

**Good diffusion model is still non-uniform $\mathsf{TC}^0$.** Consider if the score function is not perfect. Here, the the accumulated error grows slowly with $T$, i.e., $\epsilon_{\text{score}}\sqrt{\log T}$. As $T$ increases, the term $\frac{d_\epsilon(X_n)}{T}(\log T)^3$ decreases quickly, while the term $\epsilon_{\text{score}}\sqrt{\log T}$ increases only slowly and eventually dominates, setting a floor on the total error. If the error is not too large, we can still find a constant $T$ to obtain a constant error bound.

Similar interpretations can be obtained for non-uniform $\mathsf{TC}^k$ if $d_{\epsilon_n}(X_n) = O((\log n)^k)$.

There are two scenarios that the theorem's assumptions are violated and the conclusion of the theorem does not apply.

$\epsilon_{\text{score}}$ **is too large.** In this case the inequality fails and our theorem no longer applies. This corresponds to a diffusion model that is not doing a good job at its intended task of score matching. We expect this to occur when the diffusion model is poorly trained or under-parameterized, or when the network is not being used as a score-matching diffusion model at all. If it is not used for score matching, SMLD can in principle implement arbitrary Turing computations via a highly non–score-matching network.

**The intrinsic dimension of the solution space $d_\epsilon(X_n)$ grows faster than $\text{polylog}(n)$.** In this case, the convergence rate of the diffusion model will be slow enough that the task *may* go beyond non-uniform $\mathsf{TC}^k$ if $d_{\epsilon_n}(X_n) = O((\log n)^k)$. However, it doesn't immediately imply that the task is in $\mathsf{P} \setminus \mathsf{TC}$, nor the diffusion model can solve an inherently serial problem. It only means that diffusion models solve a problem with polynomial number of intrinsic dimensions with polynomial steps.

## G  INHERENTLY SERIAL PROBLEMS IN RL

Throughout this section, by a parallel algorithm, we mean specifically an L-uniform $\mathsf{TC}$ Boolean circuit family—as usual throughout this paper.

In this section, we begin with a problem from computational complexity theory that is proven to be impossible to parallelize (assuming, as always, that $\mathsf{TC} \neq \mathsf{P}$), then convert it into a deterministic decision problem. We then prove a theorem, showing that any parallel decision rule for this problem has arbitrarily bad worst-case performance. As special cases, this includes maximizing parallel value functions, maximizing parallel Q-functions, parallel policies, and parallel learning rules that produce parallel policies.

### G.1  DEFINITIONS

A Boolean circuit $C$ is **alternating** when it consists solely of AND and OR gates such that every AND gate connects to only OR gates and every OR gate connects to only AND gates.

Alternating circuits are **monotonic** in the following sense. Consider an alternating circuit that takes $n$ inputs, and let $x, x' \in \{0,1\}^n$ be two possible inputs to it. If $x \leq x'$ (i.e., $x'$ is obtained by flipping some zeros of $x$ to ones), then $C(x) \leq C(x')$. More generally, for any gate output $C_i$ of $C$, we have $C_i(x) \leq C_i(x')$; intuitively, this can be visualized as "hot" wires carrying $\texttt{True}$-signals monotonically "upwards" through the circuit.

The **depth** of a gate is the length of the longest directed path from any circuit input to that gate. A chain of the form $\text{OR} \to \text{AND} \to \text{OR} \to \cdots$ therefore has gate depths $1, 2, 3, \ldots$ in order.

For an alternating circuit $C$ on input $x$, the **depth-of-1** (DO1) of this circuit configuration is

$$d_1(C, x) := \max\{\text{depth}(g) : g \text{ outputs } 1 \text{ on } x\}.$$

The problem of computing the DO1 of any circuit configuration is the **DO1 problem**. Assuming that $\mathsf{TC} \neq \mathsf{P}$, as usual in the paper, then the DO1 problem cannot be solved by a parallel algorithm. In fact, much more can be said. Not only would it be non-parallelizable, any approximation of it is also non-parallelizable.

Fix constants $0 < \epsilon < b$. Given $(C, x)$, the problem of computing a number $d(C, x)$ satisfying

$$d(C, x) \in [\epsilon\, d_1(C, x),\ b\, d_1(C, x)].$$

Because a solution for $[\epsilon, b]$ yields one for $[\epsilon/b, 1]$, it suffices to consider the special case $b = 1$, which we call the $\epsilon$-**approximate DO1** problem.

## G.2  NON-PARALLELIZABILITY RESULTS

We quote the following result from Kirousis and Spirakis (Kirousis & Spirakis, 1988).

**Theorem G.1.** *For every $\epsilon \in (0, 1)$, the $\epsilon$-approximate DO1 problem is* P*-complete. Consequently, if* TC $\neq$ P*, no parallel algorithm can solve the $\epsilon$-approximate DO1.*

To import the theorem from computational complexity theory into RL theory, we need to construct a specific decision environment in which an agent must perform actions to maximize rewards.

The idea of the following construction is that an approximately optimal agent can be exploited as a problem-solving resource for other ends. Specifically, we will construct some environments in which the first action of the agent is a forced choice between two circuits. In the language of psychologists, we construct two-alternative forced choice experiments. The agent's choice can then be interpreted as an agent's judgment as to which circuit has a deeper depth-of-1.

Given a circuit configuration $(C, x)$, we construct, in parallel, many other circuit configurations $(C'_1, 1), (C'_2, 1), (C'_3, 1), \ldots, (C'_{|C|}, 1)$, such that $d_1(C'_1, 1) = 1, d_1(C'_2, 1) = 2, d_1(C'_3, 1) = 3, \ldots, d_1(C'_{|C|}, 1) = |C|$. Then, we perform in parallel all forced choices for these pairs: $(C, x)$ vs $(C'_1, 1)$, $(C, x)$ vs $(C'_2, 1)$, ..., $(C, x)$ vs $(C'_{|C|}, 1)$. At some point, the agent's forced binary choice should switch from preferring $(C'_k, 1)$ to preferring $(C, x)$. This can be taken as a judgment that $d_1(C, x) \approx k$. If the agent is approximately optimal, then $d_1(C, x) \approx k$ is correct, in the sense that we can guarantee $k \in [\epsilon d_1(C, x), b d_1(C, x)]$ for some constants $0 < \epsilon < b$ that do not depend on either $C$ or $x$.

This is the essential idea of the construction and the subsequent proof. The rest are tedious details designed to close loopholes.

We define the **DO1 environment** as follows. Given a circuit configuration $(C, x)$, define a corresponding deterministic decision problem as follows.

1. At $t = 0$, the agent observes two circuit configurations: the original $(C, x)$ and a length-$k$ alternating chain of the form OR$\to$AND$\to$OR$\to \cdots$, whose input is 1.

2. The agent selects one gate from one of the circuits; the unchosen one is then removed. This is the two-alternative forced choice.

3. Subsequently, the agent may select at most one additional gate per time-step, for $H := \max\{k, |C|\}$ steps, designed so that the agent has enough time to choose every desired gate.

4. The episode ends at $t = H$. If every chosen gate outputs 1, then the reward at this step is $r_H =$ the maximal depth among all the chosen gates. Otherwise, $r_H = 0$. The reward at all other steps is zero.

This means that each DO1 environment can have the following states:

- No gates are selected.
- Some gates of $C$ are selected.
- Some gates of the alternating chain are selected.

**Theorem G.2** (optimal decision in the DO1 environment is inherently serial). *Assume* TC $\neq$ P.

(a) *No parallel algorithm can compute an approximate optimal value function $V$ such that $\exists 0 < \epsilon < b$ with $V(s) \in [\epsilon V^*(s), b V^*(s)]$ for every state $s$ in every DO1 environment.*

(b) *For any parallel algorithm producing a value function $V$ and any $\epsilon \in (0, 1)$, there exists a DO1 environment where the greedy policy $\pi_V(s) := \arg\max_a V(s')$ achieves terminal reward $r_H < \epsilon r^*$, where $r^*$ is optimal. Here, $s'$ denotes the next state if, at state $s$, the agent performs action $a$.*

*(c) Statement (b) extends to any parallel algorithm that, given a state in a DO1 environment, outputs an action.*

*Proof.*    (a) We argue by contradiction. Assume there exists a parallel algorithm that, for some constant $0 < \epsilon < 1$, outputs a value function $V$ satisfying

$$V(s) \in [\epsilon\, V^*(s),\, V^*(s)] \quad \text{for every state } s \text{ in every DO1 environment.}$$

Since any factor $b > 1$ can be removed by division, we set $b = 1$ without loss of generality.

**Special case.** We need to check first the special case where $d_1(C, x) = 0$. This is done as follows. First, perform a topological sort of $C$, which is parallelizable (in $\mathsf{TC}^2$, in fact (Cook, 1985)). This then allows us to find all the gates that are in the first layer. Next, for each gate in the first layer, we test in parallel whether that gate outputs 1. Let their outputs be $y_1, \ldots, y_m$. If $\mathrm{OR}(y_1, \ldots, y_m) = 0$, then $d_1(C, x) = 0$, because alternating circuits are monotonic. Otherwise, $d_1(C, x) \geq 1$.

Having thus handled the special case, we assume that $d_1(C, x) \geq 1$ for the rest of the proof.

**Two-alternative forced choice experiments in parallel.** Given an input $(C, x)$ with $n := |C|$ gates and depth-of-1 equal to $d_1 = d_1(C, x)$, choose $m \in \mathbb{N}$ such that $2^{m-1} < n \leq 2^m$. For each index $\ell \in \{0, \ldots, m\}$ construct, in parallel, a DO1 environment $E_\ell$ whose second circuit is an alternating chain of length $2^\ell$, with the only input being `1`. Within each $E_\ell$ evaluate (again in parallel)

- $V\big(s_g^{(\ell)}\big)$, where $s_g^{(\ell)}$ is the state after initially selecting gate $g \in C$;
- $V\big(s_{\text{chain}}^{(\ell)}\big)$, where $s_{\text{chain}}^{(\ell)}$ is the state after selecting the input gate of the chain.

The whole procedure requires

$$(n + 1)(m + 1) = \big(|C| + 1\big)\big\lceil \log_2 |C| \big\rceil$$

parallel evaluations.

**Identifying the switchover index.** For $\ell = m$ the chain depth $2^m$ is at least as long as what depth-of-1 that $C$ can create, hence, if the value function is any good, it should satisfy $V\big(s_{\text{chain}}^{(m)}\big) \geq V\big(s_g^{(m)}\big)$ for all $g$. As $\ell$ decreases the chain shortens; eventually picking $C$ should be better than picking the chain. This intuition allows us to define the following decision procedure.

Let $k$ be the smallest $\ell$ such that

$$V\big(s_{\text{chain}}^{(\ell)}\big) \ \leq \ \max_g V\big(s_g^{(\ell)}\big).$$

Output $d' := 2^{m-k}$ as the estimate for $d_1$.

There are two special cases to handle. If the agent rejects the chain even for $\ell = 0$, then output $|C|$. If the agent always picks the chain, then output $d' = 1$.

**Quality of the estimate.** Suppose the switchover occurs, such that the agent picks the chain for $E_k$, but switches to picking the circuit for $E_{k-1}$. Consider in detail what happens for $E_k$.

Choose $g^*$ attaining the maximum in the definition of $k$ and set

$$s := s_{g^*}^{(k)}, \quad s' := s_{\text{chain}}^{(k)}$$

by construction, $V(s) \geq V(s')$, and by the assumed guarantee on $V$,

$$V(s) \in [\epsilon d_1,\, d_1], \qquad V(s') \in [\epsilon d',\, d'].$$

Hence $d_1 \geq \epsilon d'$.

Similarly, the argument in the case for $E_{k-1}$ shows $\epsilon d_1 \leq 2d'$, so altogether

$$d_1 \in \left[\epsilon d', \tfrac{2}{\epsilon} d'\right].$$

A similar argument applies for the two special cases where no switchover occurs.

Therefore the procedure is an approximation algorithm for DO1 that operates in parallel, contradicting Theorem G.1.

(b) A special case of (c).

(c) Assume for contradiction that there exists a parallel decision algorithm that is within an $\epsilon$ of optimality. That is, when the decision algorithm is applied in any DO1 environment, it always achieves reward at least $\epsilon r^*$. Then, for any particular DO1 environment, its first action must be an approximate solution as to whether $(C, x)$ or the chain possesses larger depth-of-1. Now, the same construction of (a) yields a parallel algorithm solving $\epsilon'$-approximate DO1, again contradicting Theorem G.1.

$\square$

**Comment.** By prepending beneath the circuit $C$ a length-$|C|$ alternating chain of gates, and making some minor adjustments to the proof, we can show that any parallel decision algorithm achieves linear regret in the worst case, meaning that $r_H - r^* = \Theta(H)$.

Although our analysis uses the DO1 problem in particular, there are some other P-complete problems, such as linear programming, that are P-complete to approximate. See for instance (Sahni & Gonzalez, 1976; Serna, 1991; Díaz et al., 1997; Greenlaw et al., 1995) for some examples. This leads us to conjecture that inherent seriality may be a fairly common phenomenon in RL, not particular to the DO1 setup.

The barrier in Theorem G.2 can be bypassed in several ways:

- In the unlikely case that $\mathsf{TC} = \mathsf{P}$ is proven, it would have great consequences for complexity theory in general, analogous to the case where $\mathsf{P} = \mathsf{NP}$ is proven. This rejects the fundamental assumption in the theorem.

- By employing a serial learning algorithm that runs in polynomial (not polylog) time, one may discover the right policy, which can then be compressed down into a policy that runs faster than serial. This bypasses the "L-uniform" part of the obstacle.

- By allowing the learned policy or value function to be non-parallel, the "TC circuit family" part of the obstacle is bypassed. This is true for certain RL algorithms, especially model-based methods that simulate the environment dynamics before making a decision.

- For some RL applications, one may be able to tolerate arbitrarily poor worst-case performance, as long as they occur rarely.

Part (b) of the theorem may explain the observation in Kevin et al. (2025). In that work, they trained both policy networks and value function networks by actor-critic methods, and noted that using deeper networks on both policy and value function improved performance. Part (b) of the theorem suggests that, when the network for approximating the value function contains less serial compute than the environment demands, then the corresponding exact $V$-maximizing policy would suffer arbitrarily bad worst-case performance. Though the theorem does not exactly apply when the policy is inexact, it suggests that the same phenomenon happens for an actual policy network trained to merely approximate the $V$-maximizing policy.

