# OpenReview forum: "The Serial Scaling Hypothesis"
_ICLR.cc/2026/Conference — ICLR 2026 Poster_

### Official Review · Reviewer_U8Ci · 2025-10-28

**Soundness:** 3
**Presentation:** 4
**Contribution:** 3
**Rating:** 8
**Confidence:** 3

**Summary:**

This paper examines the limitations of parallel scaling in modern machine learning algorithms, with a particular focus on diffusion models. The authors advocate the necessity of serial computation to overcome these limits, through the titular 'serial scaling hypothesis'. Through various examples they argue that many ML tasks require moving out of the $\textsf{TC}^0$ complexity class. The paper concludes with a theorem arguing that diffusion models with $\textsf{TC}^0$ backbones remain within $\textsf{TC}^0$, even with infinite sampling steps.

**Strengths:**

This paper is a little outside my main area of expertise. I suspect I was assigned to review this due to my experience with diffusion models, so I will focus most of my comments around that area. I have made an effort to understand other parts but it is very possible that there are aspects of the paper that I did not absorb. Overall, I noticed the following strengths:

1. The paper is well written and the authors make a compelling case in favor of serial compute capabilities in the next epoch of ML models.

2. They clearly state the limitations and assumptions behind their statements, and clarify several misconceptions that arise when talking about serial vs. parallel compute in ML. In particular they address how an architecture that is parallel, when paired with an appropriate inference procedure, can capture inherently serial computations (Sec. 2.3). I found this very helpful when trying to understand their theorem on diffusion models. I had in mind a video diffusion model which is rigged to predict the $N+1$-th frame given the first $N$ frames. Such a model can, in principle, be configured in a sliding window manner to auto regressively continue the video. If the video shows a many-body physics problem (an example from the paper) then the whole setup solves a serial problem. If I understood correctly, theorem 4.1 in the paper applies to only one of these steps, where a single frame is predicted. The latter is independent of $N$.

**Weaknesses:**

1. The proof of theorem 4.1 assumes that scores are known perfectly, which is never the case in a practical diffusion model. While the authors acknowledge this limitation in the comment at the end of App. F.2, and around line 485, they leave the practical case for future work. But this does undercut their contribution somewhat, since it is highlighted as the 'main theorem' in the paper, and since the limitations of approximate scores are central to all realistic diffusion models. Is there an opportunity here to add some experiments to bolster the claim that a generalized version of the main theorem applies if $f_\theta$ is good enough? At the very least, when speaking about the main theorem in the main text, it could be stated that it applies in an idealized limit.

2. The proof also relies on non-uniform circuit complexity, which is strictly stronger than the usual uniform classes most ML models implicitly correspond to. The former allows for a magic constant to be stored in each circuit. Since these advice strings need not be efficiently derivable or implementable in practice, it is not clear to what extent this result constrains the computational behavior of real diffusion models. A brief discussion clarifying the practical implications of this non-uniformity assumption would strengthen the claims.

3. The paper initially suggests diffusion is sequential, then proves that its step-structure provides no serial scaling. I think this is conflating serial in auxiliary diffusion time (the iterative denoising schedule) with serial in data-dependent logical dependencies (the kind of seriality complexity theory cares about). Although the authors eventually clarify that they are talking about the latter, it felt like they were conflating the two to setup the premise.

Overall, the conceptual contribution of the paper feels timely, and with some clarification around Theorem 4.1, I think the paper could be strengthened further.

**Questions:**

Small typo around line 474: 'which', not 'whiche'.

---

> ### Comment · Reviewer_U8Ci · 2025-11-14
> **Follow up**
>
> I had a follow-up question: does your theorem about diffusion models extend to discrete diffusion? That would make it even more compelling given the current trends in diffusion language modeling. Feel free to respond at your convenience.

---

> ### Author Response · Authors · 2025-12-03
>
> We appreciate the reviewer’s encouraging remarks—noting that “the authors make a compelling case in favor of serial compute capabilities in the next epoch of ML models” and that the “contribution of the paper feels timely”—as well as the thoughtful concerns and suggestions, which help push our work toward having greater impact.
>
> **Weakness 1: The proof of theorem 4.1 assumes that scores are known perfectly, which is never the case in a practical diffusion model. While the authors acknowledge this limitation in the comment at the end of App. F.2, and around line 485, they leave the practical case for future work. But this does undercut their contribution somewhat, since it is highlighted as the 'main theorem' in the paper, and since the limitations of approximate scores are central to all realistic diffusion models. Is there an opportunity here to add some experiments to bolster the claim that a generalized version of the main theorem applies if f_theta is good enough? At the very least, when speaking about the main theorem in the main text, it could be stated that it applies in an idealized limit.**
>
>
> Thanks to your encouragement. In the revised version, we now extend it to the practically relevant case of approximate score functions. The generalized statement appears as **Theorem F.2** (and is interpreted in Section F.5); we summarize it here.
>
> Let $d_\varepsilon(X_n)$ denote the intrinsic dimension of the solution manifold, and let $\varepsilon_{\text{score},n}$ be the average score error of the learned score network. The new theorem F.2 shows that for constant $c > 0$, if there exists $\delta > 0$ such that we choose the number of diffusion steps as follows: constant $T$ for constant $d_\varepsilon$, and $T_n = \mathrm{polylog}(n)$ for $d_\varepsilon(X_n) = \mathrm{polylog}(n)$, and
> $$
> c \left(\frac{d_\varepsilon(X_n)}{T}(\log T)^3 + \varepsilon_{\text{score}, n} \sqrt{\log T}\right) \leq \tfrac{1}{2} - \delta,
> $$
> then the corresponding sequence-modeling task lies in non-uniform $\mathsf{TC}^0$ (and more generally in non-uniform $\mathsf{TC}^k$ when $d_\varepsilon(X_n) = O((\log n)^k)$).
>
> For constant intrinsic dimension $d_\varepsilon(X_n) = O(1)$, this gives two useful interpretations:
>
> **Perfect diffusion model (idealized case).**
> If the score is perfect, $\varepsilon_{\text{score},n} = 0$, the condition reduces to
> $$
> c \left( \frac{d_\varepsilon(X_n)}{T} (\log T)^3 \right) \le \tfrac{1}{2} - \delta.
> $$
> We can then take $T = O(d_\varepsilon(X_n)) = O(1)$, so the task is in non-uniform $\mathsf{TC}^0$. This recovers the idealized result in the old theorem.
>
> **Good (but imperfect) diffusion model.**
> When the score is only approximate, the error accumulates as $\varepsilon_{\text{score},n}\sqrt{\log T}$. As $T$ increases, the term $\frac{d_\varepsilon(X_n)}{T}(\log T)^3$ decreases quickly, until it is limited by the slowly growing floor given by $\varepsilon_{\text{score},n}\sqrt{\log T}$. As long as $\varepsilon_{\text{score},n}$ is not too large, we can still choose a constant $T$ so that the overall bound remains below $1/2 - \delta$, and the task remains in non-uniform $\mathsf{TC}^0$.
>
> The theorem also clarifies when its conclusion does not apply:
>
> **Score error is too large.**
> If $\varepsilon_{\text{score},n}$ is too large, the inequality fails and our theorem no longer applies. Intuitively, this corresponds to a diffusion model that is not doing a good job at the task it was trained for—score matching. Our results show that the better the model approximates the true score, the “less serial” the overall computation becomes. Only if one abandons score matching altogether can one freely embed arbitrary Turing computations.
>
> **Intrinsic dimension grows too fast.**
> If the intrinsic dimension $d_\varepsilon(X_n)$ grows faster than any $\mathrm{polylog}(n)$, then we can no longer place the task in non-uniform $\mathsf{TC}^k$ via this argument. This does not mean the task is in $\mathsf{P} \setminus \mathsf{TC}$, nor that diffusion can solve an inherently serial problem; it simply means that our sufficient condition fails, and the theorem cannot be used to classify the task.

---

> > ### Author Response · Authors · 2025-12-03
> >
> > **Weakness 2: The proof also relies on non-uniform circuit complexity, which is strictly stronger than the usual uniform classes most ML models implicitly correspond to. The former allows for a magic constant to be stored in each circuit. Since these advice strings need not be efficiently derivable or implementable in practice, it is not clear to what extent this result constrains the computational behavior of real diffusion models. A brief discussion clarifying the practical implications of this non-uniformity assumption would strengthen the claims.**
> >
> >
> > Non-uniformity is essentially unavoidable due to diffusion models being a probabilistic model.
> >
> > There are only a few standard ways to deal with probabilistic models in theory:
> >
> > 1. **Remove true randomness using a non-uniformity argument.**
> >    This is what we do in the paper: we use a counting argument to fix the randomness and encode it as non-uniform “advice”.
> >
> > 2. **Remove true randomness via explicit pseudorandomness.**
> >    One could try to understand in detail which randomness is actually used in the theory, and then replace it with a carefully designed pseudorandom sequence. In our setting (and in most ML-relevant complexity arguments) this is not realistically viable.
> >
> > 3. **Keep randomness and weaken the conclusion.**
> >    Instead of an impossibility theorem, one proves only that some “bad” event happens with small probability over the randomness.
> >
> > So where does non-uniformity come from? Conceptually, it comes from the probabilistic method, which is built into modern machine learning. With “Good Old-Fashioned AI”, you first solve the problem exactly on paper, with a proof that it works on all inputs, and then you code that algorithm. In modern ML, you typically do not have such a closed-form solution; instead, you roll the dice and train a model that, with high probability will work.
> >
> > Once randomness enters the construction, the standard way to turn “with high probability” into a clean worst-case statement is to derandomize non-uniformly: by a counting argument, there exists a fixed choice of randomness that works, and we hardwire that choice.
> >
> > Since the first submission, we have also studied the case of gradient flow under the score function, where there is no further randomness once the initial point is chosen. Even here, non-uniformity reappears in the choice of initial point. The behavior of gradient flow can depend delicately on the starting point, and pathological initial points exist—for example, non-attracting stationary points (saddles) of the score field. One can make the dependence on the initial point uniform only under additional regularity assumptions that rule out such pathologies. In general, these assumptions cannot be guaranteed a priori, so non-uniformity cannot be removed in full generality.
> >
> > **Followup: Does your theorem about diffusion models extend to discrete diffusion? That would make it even more compelling given the current trends in diffusion language modeling.**
> >
> > Please refer to the common concern.

---

### Official Review · Reviewer_LMw7 · 2025-10-29

**Soundness:** 3
**Presentation:** 4
**Contribution:** 2
**Rating:** 4
**Confidence:** 4

**Summary:**

The paper argues that some problems are inherently serial and cannot be efficiently parallelized, identifying a conceptual blind spot in current scaling approaches in machine learning. The authors formalize this distinction through computational complexity theory and apply it to modern ML architectures, contending that the diffusion models cannot effectively handle inherently serial tasks. They support this argument by linking theoretical limits on parallelism with empirical patterns observed in reasoning, simulation, and decision-making tasks.

**Strengths:**

1) Raises a meaningful point: not all tasks yield to parallel scaling.
2) Connects established computational theory to machine-learning practice.
3) The theoretical framing is clear and potentially useful.
4) Adds to the discussion on the limits of diffusion and transformer architectures.
5) Writing and diagrams were very clear.

**Weaknesses:**

1) Novelty is limited; the serial/parallel distinction is long-established in complexity theory.
2) Lacks new experiments or empirical demonstrations.
3) The implications for model or hardware design remain abstract.
4) The critique of diffusion models is not representative of how text diffusion models are commonly trained today.

**Questions:**

1) Can you extend your theory to diffusion models that are trained with auto-regressive objectives or structurally autoregressive generation structures, such as block diffusion? This is a very important point to address for the serial scaling hypothesis to have practical implications for diffusion models in architectural design.
2) Comment on the translation of the work to papers like https://github.com/DreamLM/Dream-Coder
3) What concrete architectural or hardware implications follow from your theory?

I enjoyed the paper as a whole, but I can't recommend accepting unless weakness (4) and question (1) are addressed. This is my main issue with the work.

---

> ### Author Response · Authors · 2025-12-03
>
> We are grateful for the reviewer’s recognition that our work “raises a meaningful point: not all tasks yield to parallel scaling” and that the theoretical framing is “clear and potentially useful.” We also appreciate their push to examine text diffusion models in more depth, which has directly helped us sharpen and broaden the impact of the paper.
>
>
> **Weakness 1: Novelty is limited; the serial/parallel distinction is long-established in complexity theory.**
>
> We agree that the serial/parallel distinction is classical in complexity theory. Our contribution is not to reintroduce it, but to bring this viewpoint into modern ML practice and make its consequences for architectures and scaling explicit. The fact that this distinction has been known for four decades (since Cook et al. 1983) yet is still rarely used in mainstream ML is, in our view, precisely why such a bridge is needed.
>
> The **Serial Scaling Hypothesis** is intended to fill this gap by linking classical serial/parallel results to modern questions in model design and scaling, providing a framework for which approaches can handle harder tasks, and highlighting limitations of widely used architectures, including diffusion models.
>
> **The ML community often treats serial and parallel compute as interchangeable**
>
> Although Cook et al. (1983) and subsequent work have discussed inherently serial problems for decades, this perspective has not meaningfully influenced mainstream deep learning practice. For instance, Kaplan et al. (2020) introduce compute scaling laws without distinguishing serial from parallel FLOPs, and Snell et al. (2024) acknowledge two modes of scaling without emphasizing their fundamental differences. This is understandable: large text corpora mix serial and non-serial problems, making the distinction difficult to observe empirically without a guiding framework. Our work makes this distinction explicit and ties it to concrete modeling and scaling choices.
>
> **Past architectural transitions and limits of parallelization**
>
> Several historical developments suggest that the serial/parallel distinction has been repeatedly downplayed and then rediscovered. The move from RNNs (serial) to Transformers (parallel) in 2017 was largely framed as an efficiency and performance improvement; only later did works such as Merrill et al. (2023) and Chiang (2024) make explicit that highly parallel models struggle with state-tracking problems. Likewise, state-space models (linear RNNs, Mamba, etc.) initially appeared attractive because they resembled serial RNNs, but follow-up work showed that they inherit essentially the same parallelizability constraints.
>
> Attempts to parallelize RNNs show a similar pattern: many methods obtain useful speedups in certain regimes, but near the edge of stability parallelization often becomes ineffective or even slower than sequential execution. Gonzalez et al. (2024) explicitly raises this as an open question: “fundamental limitations to the computational benefit of parallelization” in these regimes. From our perspective, such observations are consistent with the conjecture \(P \neq \mathrm{TC}\): certain serial computations cannot be efficiently parallelized. The Serial Scaling Hypothesis makes this connection explicit and explains why these empirical limitations should not be surprising.
>
> **How this perspective helps with increasingly complex tasks**
>
> Modern ML challenges require not only more computation, but the right *form* of computation. As we discuss in the paper, the serial–parallel split and the associated exponential depth–width tradeoff (Lines 147–157) illustrate why genuinely serial computation becomes important for harder reasoning tasks.
>
> Today’s LLMs are fundamentally parallel models unless they are augmented with additional serial structure, such as CoT. However, it is not the only way to introduce serial computation. As summarized in Table 1, repeating layers, adding recurrency, or related architectural changes can all increase effective serial depth. Recent successes of looped Transformers (Zhu et al., 2025) and hierarchical reasoning models (Wang et al., 2025) are examples where adding serial structure yields measurable gains on more challenging tasks. The Serial Scaling Hypothesis offers a unifying explanation for these phenomena: these techniques help because they expand the model’s serial compute capacity, whereas mechanisms that remain in the purely parallel class—such as fixed-depth feedforward architectures or standard diffusion inference—face intrinsic limitations on such tasks.
>
> **References:**
>
> [Cook 1983] Cook, Stephen A. 1983. “The Classification of Problems Which Have Fast Parallel Algorithms.”
>
> [Gonzalez 2024] Gonzalez, Xavier et al. 2024. “Towards Scalable and Stable Parallelization of Nonlinear RNNs.”
>
> [Zhu 2025] Zhu, Rui-Jie et al. 2025. “Scaling Latent Reasoning via Looped Language Models.”

---

> > ### Author Response · Authors · 2025-12-03
> >
> > **Weakness 4, Question 1: The critique of diffusion models is not representative of how text diffusion models are commonly trained today. Can you extend your theory to diffusion models that are trained with auto-regressive objectives or structurally autoregressive generation structures, such as block diffusion? This is a very important point to address for the serial scaling hypothesis to have practical implications for diffusion models in architectural design.**
> >
> >
> > Please refer to the common concern.
> >
> > **Question 3: What concrete architectural or hardware implications follow from your theory?**
> >
> > In our manuscript we intentionally kept our hardware discussion brief, because giving detailed prescriptions would be somewhat premature and presumptive. We are not hardware designers, after all. Nonetheless, our theory does provide a clear hardware implication: because of the fundamental distinction between serial and parallel FLOPs, optimizing for serial tasks requires hardware that is optimized for **per-step latency**, not just overall throughput.
> >
> > **This much we are certain about.**
> >
> > If we are allowed to speculate a bit more, our educated guess is that optimizing latency will require increasing GPU clock speed, improving memory access latency, or both.
> >
> > On the clock-speed side, typical Nvidia GPUs have a clock rate below $2\,\text{GHz}$ (A100, H100, B200), which is roughly $2.5\times$ lower than a typical Intel CPU ($\sim 5\,\text{GHz}$). Our theory suggests that, for reasoning models that are bottlenecked by serial computation, it could be worthwhile to give up some peak parallel throughput in exchange for lower end-to-end latency, even if this is not an attractive choice for one-pass models such as large vision Transformers, which are primarily bottlenecked by parallel computation.
> >
> > In addition to the compute bottleneck, it has been observed that memory bandwidth poses a major bottleneck for modern AI workloads, often referred to as the “memory wall’’ (Gholami et al., 2024). When compute and memory are placed on separate chips, all communication between them has to pass through a relatively narrow perimeter interface. As a result, bandwidth scales more slowly than capacity: memory capacity can grow with the area of the chip, but off-chip bandwidth is constrained by how many wires can be brought out along the perimeter. The natural mitigation is to co-locate or distribute compute and memory, effectively increasing the communication surface between them.
> >
> > One existing example is Cerebras’ Wafer-Scale Engine (WSE), where compute units and SRAM are distributed together and connected by a fast on-wafer interconnect. The aggregate on-chip memory bandwidth is on the order of $20\,\text{PB/s}$, compared to TB/s-scale HBM bandwidth on GPUs, and this appears to translate into roughly $3$–$4\times$ higher tokens-per-second on certain large-model benchmarks compared to Nvidia’s B200 (Cerebras, 2024a; Cerebras, 2024b). We do not claim that this is the only or the “right’’ way to reduce latency, but it is an existence proof that architectures which aggressively increase on-chip memory bandwidth and reduce data movement can yield substantial gains for the kinds of workloads our theory focuses on.
> >
> > **References**
> >
> > [Gholami 2024] Gholami, Amir, Zhewei Yao, Sehoon Kim, Coleman Hooper, Michael W. Mahoney, and Kurt Keutzer. 2024. “AI and Memory Wall.” IEEE Micro 44 (3): 33–39.
> >
> > [Kang 2021] Kang, Hongbo, Phillip B. Gibbons, Guy E. Blelloch, Laxman Dhulipala, Yan Gu, and Charles McGuffey. 2021. “The Processing-in-Memory Model.” In Proceedings of the 33rd ACM Symposium on Parallelism in Algorithms and Architectures (SPAA ’21), 295–306. New York, NY: Association for Computing Machinery.
> >
> > [Cerebras 2025a] Cerebras Systems. 2025. “Cerebras CS-3 vs Nvidia DGX B200 Blackwell.” Cerebras Blog, September 19, 2025. Available at: https://www.cerebras.ai/blog/cerebras-cs-3-vs-nvidia-dgx-b200-blackwell.
> >
> > [Cerebras 2025b] Cerebras Systems. 2025. “Cerebras CS-3 vs Groq LPU.” Cerebras Blog, September 19, 2025. Available at: https://www.cerebras.ai/blog/cerebras-cs-3-vs-groq-lpu.

---

### Official Review · Reviewer_wmHT · 2025-10-31

**Soundness:** 3
**Presentation:** 4
**Contribution:** 3
**Rating:** 8
**Confidence:** 5

**Summary:**

The paper reveals that modern machine learning overlooks problems that are fundamentally sequential and cannot be parallelized. It formalizes this distinction in complexity theory, showing that parallel architectures face theoretical limitations on such tasks. The study further shows that diffusion models also fail on inherently serial problems.

**Strengths:**

- The writing is concise and sharp and the study focuses on a precise and underexplored question: the boundary between parallel and inherently serial computation.
- Examples from reasoning, physics, and decision-making vividly illustrate what “inherently serial” means. Theoretical formalization combined with diffusion model evidence creates a rigorous and convincing argument.

**Weaknesses:**

- The proposed hypothesis trys to introduce computational complexity of serial dependency into model performance evaluation. However, the many-body example of serial problem is a prediction task on chaotic systems, where, even if we have ground truth labels and prior knowledge to train a model, the generalization performance is doubtfully discussed by a general hypothesis and experimental supports. This stems from the nature of the problem itself. It seems not good to use a many-body case here.
- Further numerical experiments with claim-relate analyses can validate the relevant conclusions and explore future research directions, such as how to verify the proposed hypotheses. This will make the factual existence clearer to the audience.

**Questions:**

In reasoning QA tasks, could more ablation about CoT or not be provided? The claims here try to specialize the type of QA as inherently serial, but it's hard to say that the reasoning and knowledge are decoupled and the models are actually reasoning without ablation.

---

> ### Author Response · Authors · 2025-12-03
>
> We thank the reviewer for their very positive assessment, in particular the comments that “the writing is concise and sharp” and that our “theoretical formalization combined with diffusion model evidence creates a rigorous and convincing argument.” We are encouraged by the reviewer’s high-confidence recommendation and are glad that the core question and presentation resonated clearly.
>
> **Weakness 1: The proposed hypothesis tries to introduce computational complexity of serial dependency into model performance evaluation. However, the many-body example of a serial problem is a prediction task on chaotic systems, where, even if we have ground truth labels and prior knowledge to train a model, the generalization performance is doubtfully discussed by a general hypothesis and experimental supports. This stems from the nature of the problem itself. It seems not good to use a many-body case here.**
>
> Thank you for raising this point. It’s an important distinction, and we realize the current draft doesn’t emphasize it clearly enough.
>
> To clarify: the many-body problem discussed in our manuscript is the **finite-precision** variant, which is not chaotic and is in fact P-complete. The reviewer is absolutely correct that the general many-body problem, with unbounded precision and extreme sensitivity to initial conditions, is PSPACE-hard. The hardness hinges on requiring arbitrarily high precision to faithfully track the system’s evolution.
>
> In contrast, the finite-precision setting partitions the space into a fixed number of discrete cells, and the system evolves deterministically according to the underlying physical laws. Because precision is bounded, this version does not exhibit chaotic sensitivity, and its associated prediction problem falls in P-complete, which is the complexity class relevant to our arguments about serial computation.
>
> We agree this distinction deserves more emphasis, and we will revise the final version to make it explicit. We are considering moving Footnote 3 (“Finite precision reflects practical scientific computation. With unbounded precision, related prediction problems become PSPACE-hard.”) into the main text.

---

### Author Response · Authors · 2025-12-03
**A summary for AC**

We thank the reviewers for their constructive feedback and are grateful that all three engaged positively with the paper. R1 wrote that “theoretical formalization combined with diffusion model evidence creates a rigorous and convincing argument,” R2 noted that the work “raises a meaningful point: not all tasks yield to parallel scaling” and found the framing “clear and potentially useful,” and R3 felt that “the conceptual contribution of the paper feels timely.”

We recognize that your time is limited, so below we concisely summarize the major concerns raised by the reviewers along with our brief responses.

# Major revision

Thanks to the encouragement from the reviewers, we have revised the manuscripts to make our arguments more generally applicable (Section F in the appendix (pp. 23–28), in Blue).

**Generalized diffusion theorem**

**Previously (Theorem F.1):** We showed that because diffusion models have the convergence rate of $O(d/T)$, where $d =$ number of denoising dimensions – independent of problem size $n$, it has the total computation steps as $O(1)$ and can only solve $\mathrm{TC}^0$ problems.

Two major assumptions:
1. The diffusion models are modeling a valid score function, and
2. The denoising dimensions are fixed.

**Now:** We updated our manuscript with the new **Theorem F.2** which generalizes and improves over the old theorem in two ways:
**(1) it applies to even approximate score functions**, and
**(2) it shows that the convergence rate only depends on the “intrinsic dimension” ($d_{\text{eps}}$) instead of the “denoising dimensions” ($d$)**, where $d_{\text{eps}}$ can be much smaller than $d$.

This revision is important because:

Now, the diffusion models do not need to be a valid score function, they just need to be “close enough” which is explained in detail in Section F.5.

We can now show that diffusion modeling of videos that are governed by physical laws is not serial. Because these videos tend to have very small or even constant intrinsic dimensions. Why? The more predictable the next frames are, the fewer the valid continuations, and fewer intrinsic dimensions. Since the new theorem says that the diffusion models only scale computation with the intrinsic dimensions, yet the seriality of a problem has nothing to do with the number of intrinsic dimensions. Diffusion modeling of physical videos is not serial. This might explain why video diffusion models demonstrate poor physics understanding. Read more in Section F.6.


# Major concerns

Note: full, reviewer-specific replies (including other issues) appear in the individual reviewer threads below.

**R2 W1: Novelty is limited; the serial/parallel distinction is long-established in complexity theory.**

We agree that the serial/parallel distinction is classical in complexity theory. Our contribution is not to reintroduce it, but to bring this viewpoint into modern ML practice and make its consequences for architectures and scaling explicit. The fact that this distinction has been known for four decades yet is still rarely used in mainstream ML is, in our view, precisely why such a bridge is needed: current practice and scaling laws typically treat serial and parallel FLOPs interchangeably.

There are several historical developments where this distinction was initially downplayed and only later brought back into focus: the RNN→Transformer transition (2017), the rise of state-space models (2021–23), and recent attempts to parallelize RNNs, where follow-up work highlights limitations of highly parallel models on long-horizon state-tracking and, in the RNN case, reports that near the edge of stability parallelization can be ineffective. The Serial Scaling Hypothesis highlights the conjecture $\(P \neq \mathrm{TC}\)$, suggesting that some tasks genuinely require increasing serial compute, and helps explain why methods that add serial structure often yield gains on more challenging reasoning tasks.

**R3 W1: The proof of theorem 4.1 assumes that scores are known perfectly, which is never the case in a practical diffusion model.**

The new theorem F.2 has generalized to cover the approximate score function case.

**R3 W2: The proof also relies on non-uniform circuit complexity, which is strictly stronger than the usual uniform classes most ML models implicitly correspond to.**

Non-uniformity is hard to avoid due to diffusion models being a probabilistic model. Once randomness is involved, the standard way to obtain worst-case statements is to derandomize via a counting argument, which yields non-uniform “advice.” Alternatives—replacing true randomness with explicit pseudorandomness or changing to high-probability statements—are either unrealistic in our setting or change the nature of the theorem. Even in the deterministic gradient-flow setting, non-uniformity reappears through the choice of initialization: avoiding pathological starting points is impractical in  general.

---

> ### Author Response · Authors · 2025-12-03
> **Common concerns**
>
> **R2: “The critique of diffusion models is not representative of how text diffusion models are commonly trained today. Can you extend your theory to diffusion models that are trained with auto-regressive objectives or structurally autoregressive generation structures, such as block diffusion?”**
>
> **R3: “Does your theorem about diffusion models extend to discrete diffusion? That would make it even more compelling given the current trends in diffusion language modeling.”**
>
> This is an important point, especially because text diffusion has become increasingly popular in NLP.
>
> There are two separate concerns:
> 1. Our diffusion theorems, both the original and the revised version, are stated for continuous diffusion models. They do not immediately generalize to discrete diffusion, which is what most text diffusion models use.
> 2. Text diffusion models often use chain-of-thought (CoT) or other reasoning tokens. These intermediate tokens may inflate the intrinsic dimension of the solution manifold, slowing convergence and potentially violating the assumptions of Theorem F.2.
>
> During the rebuttal period, we were not able to fully generalize our theorem to the discrete case. However, **if we allow the conjecture** that discrete diffusion has similar convergence properties as do continuous diffusion models (defined below), we can show that **text diffusion models are not serial reasoners.** After that, we demonstrate a **computational landscape** for common text diffusion models according to their seriality.
>
> **Conjecture 1.** Discrete diffusion models converge at a similar rate or faster than $O(d_v/T)$, where $d_v$ is a discrete intrinsic dimension (defined below), and the arguments of Theorem F.2 extend to discrete diffusion models.
>
> For discrete sequence problems over an alphabet of size $v$, let $S_n \subseteq [v]^{n}$ denote the set of valid sequences (the solution manifold) for length $n$. We define a **discrete intrinsic dimension** by
> $$
> d_v(X_n) \coloneqq \log_v |S_n|.
> $$
> Intuitively, $|S_n| \approx v^{d_v(X_n)}$ means that the solution manifold occupies the same combinatorial “volume’’ as a $d_v(X_n)$-dimensional subset of the ambient $n$-dimensional grid. In particular, if $|S_n| = O(1)$, then $d_v(X_n) = O(1)$; if $|S_n|$ is polynomial in $n$, then $d_v(X_n) = O(\log n)$; whereas an exponentially large solution set $|S_n| = v^{\Theta(n)}$ corresponds to $d_v(X_n) = \Theta(n)$.
>
> This definition is just one example among many possibilities, but it is sufficient for the arguments below.
>
> **Intrinsic dimension and discrete diffusion for long, logical sequences.**
> Section F.6 emphasizes that intrinsic dimension is crucial for understanding diffusion modeling of long but logically constrained sequences. There, we discussed video diffusion and argued that physical videos have very low intrinsic dimension, which helps explain why diffusion models struggle with true physical reasoning despite being able to model long sequences.
>
> For text diffusion, we consider settings with intermediate reasoning tokens before the final answer, such as CoT or cellular automata (CA) traces.
>
> **Cellular automata.**
> Each cell has two possible values, making CA a natural fit for discrete diffusion. Suppose the answer is the $N$-th row of a cellular automaton with a fixed rule and initial condition. The intermediate tokens are the $N-2$ rows between the initial row and the answer; each row contains $2N-1$ tokens, for a total of $(N-2)(2N-1)$ intermediate tokens. Although this is a very long sequence, there is only one valid continuation for a given rule and initial condition. According to the above definition, this gives $d_v(X_n) = O(1)$, so (under Conjecture 1) discrete diffusion converges quickly in a fixed number of steps. In other words, in this CA setting the discrete diffusion model is not a serial model, mirroring the argument in Section F.6.
>
> **Math QA with CoT.**
> In math QA, CoT tokens describe arithmetic operations $o_1, o_2, \ldots, o_i, \ldots$ with length $O(n)$, proportional to the problem size. Section 3.5 shows that math QA is an inherently serial problem because many operations must be performed in a specific order.
>
> If the order of all $O(n)$ operations is strict, then there is only one valid CoT, so $d_v(X_n) = O(1)$ and the same CA-like argument applies: discrete diffusion is not inherently serial under Conjecture 1.
>
> If the ordering of operations does not matter at all, then the computation can be parallelized (e.g., as a balanced tree of operations). In this case the underlying problem is in a $\mathsf{TC}$ class rather than being inherently serial.
>
> Thus some inherently serial problems (such as CA evolution to a specific time) can still have only $O(1)$ valid CoT continuations, making discrete diffusion models—which are tied to intrinsic dimension rather than sequence length—not serial in general.

---

> > ### Author Response · Authors · 2025-12-03
> > **A landscape of text diffusion models**
> >
> > “Text diffusion model” covers a wide range of architectures. In our context, three properties are relevant:
> >
> > 1. **Is the diffusion model discrete?**
> >    Our Theorem F.2 still applies to continuous diffusion on text domains, such as Dieleman et al. (2022). For fully discrete models, we rely on Conjecture 1.
> >
> > 2. **Does decoding happen left-to-right in an autoregressive (AR) manner?**
> >    Autoregressive decoding is known to be serial: if predicting each next token lies in $\mathsf{TC}^0$ (which is true given a suitable CoT), then any model that incorporates left-to-right AR decoding is itself a serial model. Block Diffusion (Arriola et al., 2025) is an example: decoding proceeds autoregressively at the block level. Interestingly, their Table 3 reports that smaller block sizes yield better perplexity and that a fully AR model performs best. This is consistent with the view that pure discrete diffusion is not inherently serial, and that serial structure reappears when AR is re-introduced.
> >
> > 3. **Is decoding confidence-based rather than position-based?**
> >    In some text diffusion models, decoding proceeds by repeatedly filling in the most confident token (or set of tokens), often for $O(n)$ steps. If each step adds one CoT token, the model effectively becomes an autoregressive model in which the autoregressive order is determined by the model’s own confidence. This is again a serial model. Examples include LLADA (Nie et al., 2025), Dream 7B and Dream-Coder (Xie et al., 2025), and Efficient DLM: all use confidence-based decoding and achieve performance close to AR language models in terms of perplexity.
> >
> > By contrast, vanilla discrete diffusion denoises all tokens equally at each step. In masked (absorbing-state) diffusion, tokens are randomly unmasked. Because the decoding order is not aligned with the serial dependencies of inherently serial tasks, the model will often decode parts of the CoT “out of order”, effectively taking shortcuts that do not exist in the true serial computation.
> >
> > ---
> >
> > ### Summary table of text diffusion models
> >
> > Below we summarize several commonly referenced text diffusion models and their properties in our framework:
> >
> > | Model                                        | Discrete? | Decoding left-to-right?     | Confidence-based decoding? | Serial without CoT?             | Serial with CoT?                      |
> > |---------------------------------------------|----------:|-----------------------------|----------------------------|----------------------------------|--------------------------------------|
> > | Dieleman et al. (2022)                      | No        | No                          | No                         | No (Theorem F.2)                 | No (with Conjecture 1)               |
> > | Hoogeboom et al. (2021)                     | Yes       | No                          | No                         | No (with Conjecture 1)           | No (with Conjecture 1)               |
> > | D3PM (Austin et al., 2021)                  | Yes       | No                          | No                         | No (with Conjecture 1)           | No (with Conjecture 1)               |
> > | SEDD (Lou et al., 2024)                     | Yes       | No                          | No                         | No (with Conjecture 1)           | No (with Conjecture 1)               |
> > | Reparam. Discrete Diffusion (Zheng, 2023)   | Yes       | No                          | Yes                        | No (with Conjecture 1)           | Yes (confidence-based decoding)      |
> > | Sahoo et al. (2024)                         | Yes       | No                          | No                         | No (with Conjecture 1)           | No (with Conjecture 1)               |
> > | Block Diffusion (Arriola et al., 2025)      | Yes       | Yes, at block level         | No                         | No (with Conjecture 1)           | Yes (block-level autoregressive decoding) |
> > | Nie et al. (2025)                           | Yes       | No                          | Yes                        | No (with Conjecture 1)           | Yes (confidence-based decoding)      |
> > | Dream-Coder (Xie et al., 2025)              | Yes       | No                          | Yes                        | No (with Conjecture 1)           | Yes (confidence-based decoding)      |
> >
> > **References:**
> >
> > [Efficient DLM] “Efficient-DLM: From Autoregressive to Diffusion Language Models, and Beyond in Speed.” 2025. In The Fourteenth International Conference on Learning Representations. https://openreview.net/forum?id=kIbFgoCq86.

---

### Meta-Review · Area_Chair_MpCC · 2025-12-16

**Summary:**

The reviewers agree that this is an interesting work that raises a valid question about the boundary between parallel and serial computations in modern machine learning problems and approaches. They also appreciate the theoretical foundation for the diffusion models. The presentation is clear, and the examples are illustrative.

**Reviewer Concerns:**

To my best understanding, most concerns have been addressed by the authors. However, as the authors acknowledge in the common concern comment, they were unable to extend their result to the practical discrete diffusion case and only leave a conjecture.

**Reviewer Scores:**

The reviewers are generally positive and recommend acceptance. However, Reviewer LMw7 evaluated the work with "Rating: 4" and asked to clarify one major weakness and question, which was not fully addressed in the common concern comment.

---

### Decision · Program_Chairs · 2026-01-26

Accept (Poster)